# Whole genome sequencing of *Neisseria meningitidis* Y isolates collected in the Czech Republic in 1993-2018

**Michal Honskus**[1,2], **Zuzana Okonji**[1‡], **Martin Musilek**[1‡], **Pavla Krizova**[1]*

**1** National Reference Laboratory for Meningococcal Infections, Centre for Epidemiology and Microbiology, National Institute of Public Health, Prague, Czech Republic, **2** Third Faculty of Medicine, Charles University, Prague, Czech Republic

☯ These authors contributed equally to this work.
‡ ZO and MM also contributed equally to this work.
* pavla.krizova@szu.cz

**Data Availability Statement:** All WGS data are available on webpage of the Neisseria PubMLST database (www.pubmlst.org/neisseria/), under the IDs presented in Table 1 of the paper.

## Abstract

### Introduction

The study presents the analysis of whole genome sequencing (WGS) data for *Neisseria meningitidis* serogroup Y isolates collected in the Czech Republic and their comparison to other countries. The aim of the study was to determine whether there are lineages of *N. meningitidis* serogroup Y in the Czech Republic genetically related to foreign ones that have been causing an increase of the morbidity and the mortality of invasive meningococcal disease (IMD) world-wide recently.

### Material and methods

The WGS data of 43 Czech *N. meningitidis* Y isolates, 35 from IMD and 8 from healthy carriers were analysed. Due to the potential of meningococcal B vaccines to induce protection against non-B serogroups, the coverage of Czech isolates of *N. meningitidis* Y by these vaccines was studied. The WGS data of Czech, European and non-European isolates of *N. meningitidis* serogroup Y were compared.

### Results

WGS assigned 36 isolates of *N. meningitidis* Y to five clonal complexes: cc23, cc92, cc167, cc103, and cc174, while seven isolates remained unassigned to any clonal complexes (ccUA). Eighteen invasive isolates belonged to clonal complex cc23, which was detected throughout the studied years. The occurrence of cc23 was recorded in all age groups of IMD patients, with the highest found in those aged 15–19 years. On the phylogenetic network isolates of cc23 form a separate lineage, distinct from all other isolates of *N. meningitidis* Y. The remaining isolates were assigned to other clonal complexes and have very low relatedness to cc23 isolates and to each other. The comparison with foreign WGS data showed that within the main genetic lineages, which are defined by clonal complexes, Czech isolates

**Funding:** Supported by Ministry of Health of the Czech Republic, grant no. NV19-09-00319. All Rights reserved. The funding had no role in study design, data collection and analysis, decision to publish, or preparation of the manuscript.

of *N. meningitidis* Y, similar to European ones, mostly cluster together and form geographical sublineages.

## Conclusions

WGS analysis showed the population of Czech *N. meningitidis* Y isolates as relatively heterogeneous, containing a large number of genetic lineages. The Czech isolates of *N. meningitidis* Y follow the trend observed for European isolates. Our result was one of the bases for updating the recommended vaccination strategy in the Czech Republic.

## Introduction

Invasive meningococcal disease (IMD) is one of the most severe infectious diseases worldwide with a high fatality rate and a high risk of life-long disabling sequelae in survivors [1–3]. IMD is caused by the bacterium *Neisseria meningitidis*, detected in the upper respiratory tract of up to 10% of the healthy population, referred to as healthy carriers of meningococci. The serogroups of *N. meningitidis* most often involved in IMD are A, B, C, X, W, and Y. Their distribution varies between countries [4–6].

The incidence of IMD and causative serogroups vary over time partly due to a combination of effects of vaccination programmes and also natural fluctuation. After meningococcal C conjugate vaccines were introduced in the early 2000s, a sharp decline in IMD caused by *N. meningitidis* serogroup C was observed in a number of countries. When meningococcal A conjugate vaccine became available in the African meningitis belt it practically led to the elimination of IMD caused by *N. meningitidis* serogroup A in this area [3]. Consequently, other causative serogroups started to be dominant. Increasing trends were observed particularly for serogroups W and Y, which resulted in switching from monovalent meningococcal C conjugate vaccines to quadrivalent meningococcal A, C, W, Y conjugate vaccines in many countries [3].

Since the 1990s IMD caused by serogroup Y showed an upward trend and increased case fatality rate in the USA, Canada, Latin American countries, but also in Europe, particularly in Sweden [6–14]. Before the worldwide rise in IMD caused by serogroup Y, this serogroup was detected primarily in healthy carriers, being rarely the cause of IMD and if so, particularly in the elderly population. The recent rise in IMD cases caused by serogroup Y has also been reported in younger age groups [13]. Molecular characterization of isolates of *N. meningitidis* Y has shown that the recent rise in IMD caused by this serogroup is primarily linked to clonal complex cc23, which, however, had a relatively low invasive potential compared to other hyperinvasive lineages [14].

In the Czech Republic, data on the incidence of IMD have been available since 1943 (passive reporting) along with surveillance data since 1993 [15]. The surveillance data-based analysis of the epidemiological situation of IMD in the Czech Republic and the meningococcal vaccination guidelines are published annually in the Bulletin of the Centre for Epidemiology and Microbiology (http://www.szu.cz/publikace/zpravy-epidemiologie-a-mikrobiologie). The IMD surveillance data in the period from 1993 to 2018 show the highest incidence 2.2/100 000 population in 1995. After this, the incidence gradually decreased reaching the minimum 0.4/100 000 population in 2014 and 2016. The IMD surveillance data from 1993 to 2018 indicate that serogroups B and C are dominant in the Czech Republic (ranging from 20.4 to 71.9% and from 6.6 to 59.2%, respectively), but since the early millennium, serogroups W and Y have

been on the rise: up to 9.3% and 5.9%, respectively (S1 Fig). The IMD surveillance programme in the Czech Republic also includes molecular characterization of isolates performed by the National Reference Laboratory for Meningococcal Infections (NRL) [15], which has been extended by the WGS method since 2015 [16–18].

Several polysaccharide conjugate vaccines against serogroups A, C, W and Y are available [19]. Two recombinant vaccines including serogroup B meningococcal antigens (MenB vaccines) have been developed: MenB-4C vaccine (Bexsero) and MenB-fHbp vaccine (Trumenba) [20, 21]. Genes encoding MenB vaccine antigens are also present in strains belonging to other meningococcal serogroups and for this reason, theoretical coverage of non-B *N. meningitidis* isolates by MenB vaccines is being studied worldwide [22–25]. Protection by MenB vaccines against non-B *N. meningitidis* represents added value in vaccination programs [26]. Vaccines registered by the European Medicines Agency (EMA) are available in the Czech Republic for vaccination against IMD–two quadrivalent conjugate vaccines: MenACWY-TT vaccine (Nimenrix) and MenACWY-CRM vaccine (Menveo) and both MenB vaccines (Bexsero, Trumenba). To achieve the most comprehensive possible immunity against IMD, the combination of a quadrivalent ACWY conjugate vaccine and MenB vaccine is recommended in the Czech Republic. In accordance with Czech legislation, vaccination against IMD is covered by health insurance for persons with underlying diseases (since January 2018) and small children (since May 2020).

Since 1993, the overall case fatality rate of IMD in the Czech Republic has been ranging from 4.7 to 16.4%, averaging at 10.2%. The highest case fatality rate is due to IMD caused by *N. meningitidis* Y averaging at 15.5%. Molecular characterization of *N. meningitidis* Y isolates provides the necessary background data for setting the vaccination strategy in the Czech Republic. We present results of the WGS analysis of 43 Czech isolates of *N. meningitidis* Y from 1993–2018 and their comparison with the WGS data from other countries. The aim of the study was to determine whether there are lineages of *N. meningitidis* serogroup Y in the Czech Republic genetically related to foreign ones that have been causing an increase of the morbidity and the mortality of invasive meningococcal disease worldwide recently.

## Material and methods

### *Neisseria meningitidis* isolates

Isolates of *N. meningitidis* from IMD patients are referred to the NRL by microbiological laboratories from all over the Czech Republic for confirmation and further characterization in accordance with the Czech legislation. The isolates from healthy carriers are referred to the NRL voluntarily and in small numbers. All isolates received are stored freeze-dried and/or frozen (-80˚C, Cryobank B, ITEST) in the NRL collection. For each isolate, clinical, epidemiological, and microbiological data are available in the electronic database of the NRL collection. A total of 43 *N. meningitidis* serogroup Y isolates from 1993–2018 were studied, 35 from IMD patients and 8 from healthy carriers, to determine if there is a difference in isolates from patients with IMD and from healthy carriers.

### Characterization of *Neisseria meningitidis*

*N. meningitidis* isolates were cultured on Mueller Hinton chocolate agar at 37˚C and 5% $CO_2$ for 18–24 hours. The identification of *N. meningitidis* was confirmed using the API NH kit (BIOMÉRIEUX). Serogroups were determined by conventional serological methods (Pastorex Meningitis Bio-RAD, antisera *N. meningitidis* ITEST, Bio-RAD) and confirmed by RT-PCR.

## Whole genome sequencing and analysis

The methods used in the present study have been described in detail previously [17, 18]. DNA isolation was performed using the QIAamp DNA Mini Kit (QIAGEN) according to the manufacturer's instructions. DNA was sent for WGS to the European Molecular Biology Laboratory, Heidelberg, Germany (EMBL). The Illumina MiSeq platform was used for sequencing and the result was overlapping sequences approximately 300 bp in length. The genome assembly from primary raw WGS data was performed using the Velvet *de novo* Assembler software with Velvet-Optimiser script [27]. Isolate genome assemblies were submitted to the PubMLST *Neisseria* database (https://pubmlst.org/) [28, 29] and automatically characterized by the BIGSdb platform at finetyping loci (*porA, fetA*) [30], MLST genes (*abcZ, adk, aroE, fumC, gdh, pdhC,* and *pgm*) [31], MenB vaccine antigen genes (*nhba, nadA,* and *fHbp*) [32] and 53 ribosomal protein genes (*rpsA–rpsU, rplA–rplF, rplI–rplX, rpmA–rpmJ*) [33]. New gene and peptide variants were scanned manually, and after curator approval and annotation, were added to the PubMLST database. Genomes were compared using the BIGSdb Genome Comparator tool with the "all loci" scheme, providing the highest resolution. Incomplete loci (due to contig breaks) were ignored in pairwise comparisons in the distance matrix calculations. The distance matrices, which are based on the number and allelic variability of the genes contained in individual schemes, were generated automatically and phylogenetic networks constructed using the SplitsTree4 software using the NeighborNet algorithm [34]. Results were graphically edited by the Inkscape tool (www.inkscape.org/en/).

Isolates were assigned to sequence type (ST), clonal complex (cc) and ribosomal sequence type (rST) based on the allelic profile of MLST and ribosomal protein genes. Due to the potential of MenB vaccines to induce protection against non-B serogroups, the coverage of isolates by these vaccines was studied. The Bexsero Antigen Sequence Type (BAST) was determined from the combination of peptide variants of two variable regions of the PorA protein (VR1 and VR2) and peptide variants of three MenB-4C vaccine antigens (NHBA, NadA, and FHbp) [35]. From this, the theoretical coverage of a given isolate by both MenB vaccines (Bexsero and Trumenba) was determined using the Meningococcal Deduced Vaccine Antigen Reactivity (MenDeVAR) Index, recently introduced on the PubMLST database. The MenDeVAR index is based on analysis of the presence of particular peptide variants of MenB vaccine antigens, their expression, and bactericidal susceptibility test result [36].

The phylogenetic networks were created for invasive isolates of *N. meningitidis* Y from the Czech Republic (n = 35) together with European (n = 960) and non-European countries (n = 375), which were available in the PubMLST database by May 26, 2021 and met the following criteria: serogroup Y, genogroup Y, invasiveness, complete MLST profile, and sequencing data size of > 1.8 MB.

## Results

### Czech isolates of *N. meningitidis* Y

WGS assigned 36 of 43 isolates of *N. meningitidis* Y to five clonal complexes: cc23, cc92, cc167, cc103, and cc174, while seven isolates remained unassigned to any of the described clonal complexes (ccUA) (Table 1). Most prevalent was cc23 with 21 isolates (ST-23, n = 10; ST-569, n = 4; ST-1625, n = 6; and ST-8526, n = 1). Followed by cc92 with 7 isolates (ST-92, n = 6, and ST-94, n = 1), and cc167 with 6 isolates (ST-168), as well as one isolate each for cc103 (ST-11017) and cc174 (ST-1466). The most frequent clonal complex in isolates from IMD was cc23, followed by cc167, while in isolates from healthy carriers, cc92 followed by cc23 was most frequently found. The highest occurrence of IMD caused by *N. meningitidis* Y in the

**Table 1. Basic molecular characterization of *N. meningitidis* Y isolates collected in the Czech Republic from 1993 to 2018, (n = 43).**

| Strain IDs | PubMLST IDs | ST | *abcZ* | *adk* | *aroE* | *fumC* | *gdh* | *pdhC* | *pgm* | cc | *porA* VR1 | *porA* VR2 | *fetA* VR | rST |
|---|---|---|---|---|---|---|---|---|---|---|---|---|---|---|
| 0059/93 | 966 | 92 | 3 | 7 | 4 | 37 | 8 | 18 | 21 | ST-92 | 5–1 | 10–14 | F5-14 | 3142 |
| 0111/93 | 1580 | 130 | 23 | 5 | 9 | 3 | 13 | 32 | 8 | UA | 18–7 | 9 | F5-13 | 3297 |
| 0224/93 | 1614 | 94 | 3 | 7 | 4 | 35 | 8 | 18 | 2 | ST-92 | 5–1 | 10–4 | F1-5 | 3214 |
| 0323/93 | 8142 | 92 | 3 | 7 | 4 | 37 | 8 | 18 | 21 | ST-92 | 5–1 | 10–1 | F5-14 | 3006 |
| 0338/93 | 8152 | 92 | 3 | 7 | 4 | 37 | 8 | 18 | 21 | ST-92 | 5–1 | 10–1 | F5-14 | 3006 |
| **0200/94** | 83899 | 92 | 3 | 7 | 4 | 37 | 8 | 18 | 21 | ST-92 | 5–1 | 10–4 | F1-5 | 137524 |
| **0040/95** | 83871 | 569 | 10 | 5 | 93 | 14 | 11 | 9 | 17 | ST-23 | 5–1 | 2–2 | F5-8 | 2422 |
| **0449/95** | 83903 | 92 | 3 | 7 | 4 | 37 | 8 | 18 | 21 | ST-92 | 5–1 | 10–4 | F1-5 | 137524 |
| **0008/96** | 83865 | 130 | 23 | 5 | 9 | 3 | 13 | 32 | 8 | UA | 12–1 | 16–8 | F5-13 | 137518 |
| **0264/99** | 83901 | 1625 | 10 | 5 | 18 | 14 | 11 | 9 | 17 | ST-23 | 5–1 | 2–2 | F5-8 | 137525 |
| **0172/01** | 83895 | 1625 | 10 | 5 | 18 | 14 | 11 | 9 | 17 | ST-23 | 5–1 | 2–2 | F5-8 | 2422 |
| **0179/01** | 83897 | 569 | 10 | 5 | 93 | 14 | 11 | 9 | 17 | ST-23 | 5–1 | 2–2 | F5-8 | 137523 |
| **0251/01** | 83900 | 23 | 10 | 5 | 18 | 9 | 11 | 9 | 17 | ST-23 | 5–2 | 10–1 | F4-1 | 2421 |
| **0055/02** | 83874 | 23 | 10 | 5 | 18 | 9 | 11 | 9 | 17 | ST-23 | 7–2 | 13–2 | F4-17 | 137520 |
| **0068/02** | 83877 | 1625 | 10 | 5 | 18 | 14 | 11 | 9 | 17 | ST-23 | 5–1 | 2–2 | F5-8 | 137521 |
| **0177/02** | 83896 | 569 | 10 | 5 | 93 | 14 | 11 | 9 | 17 | ST-23 | 5–1 | 2–2 | F5-8 | 2422 |
| **0066/03** | 83911 | 130 | 23 | 5 | 9 | 3 | 13 | 32 | 8 | UA | 18–12 | 10–2 | F2-7 | 137529 |
| 0157/03 | 83922 | 3015 | 219 | 5 | 275 | 17 | 11 | 8 | 21 | UA | 5–2 | 10–1 | F5-8 | 7234 |
| **0192/04** | 83898 | 1625 | 10 | 5 | 18 | 14 | 11 | 9 | 17 | ST-23 | 5–1 | 2–2 | F5-8 | 2422 |
| **0267/04** | 83902 | 569 | 10 | 5 | 93 | 14 | 11 | 9 | 17 | ST-23 | 5–1 | 2–2 | F5-8 | 2589 |
| **0065/05** | 83910 | 3015 | 219 | 5 | 275 | 17 | 11 | 8 | 21 | UA | 5–2 | 10–1 | F5-8 | 137528 |
| **0075/05** | 83913 | 1625 | 10 | 5 | 18 | 14 | 11 | 9 | 17 | ST-23 | 5–1 | 2–2 | F5-8 | 2422 |
| **0094/07** | 83915 | 23 | 10 | 5 | 18 | 9 | 11 | 9 | 17 | ST-23 | 5–2 | 10–1 | F4-1 | 137531 |
| **0105/07** | 83917 | 130 | 23 | 5 | 9 | 3 | 13 | 32 | 8 | UA | 18–12 | 10–2 | F2-7 | 137532 |
| **0156/07** | 83921 | 3015 | 219 | 5 | 275 | 17 | 11 | 8 | 21 | UA | 18–1 | 16–4 | F3-4 | 137534 |
| **0190/07** | 83923 | 168 | 2 | 16 | 6 | 17 | 9 | 18 | 8 | ST-167 | 5–1 | 10–4 | F4-1 | 7517 |
| **0121/08** | 83920 | 23 | 10 | 5 | 18 | 9 | 11 | 9 | 17 | ST-23 | 5–2 | 10–1 | F4-1 | 2421 |
| 0125/08 | 106031 | 23 | 10 | 5 | 18 | 9 | 11 | 9 | 17 | ST-23 | 5–2 | 10–1 | F4-1 | 2421 |
| 0126/08 | 106032 | 23 | 10 | 5 | 18 | 9 | 11 | 9 | 17 | ST-23 | 5–2 | 10–1 | F4-1 | 2421 |
| **0049/09** | 83872 | 23 | 10 | 5 | 18 | 9 | 11 | 9 | 17 | ST-23 | 5–2 | 10–1 | F4-1 | 2421 |
| **0039/10** | 83870 | 168 | 2 | 16 | 6 | 17 | 9 | 18 | 8 | ST-167 | 5–1 | 10–4 | F4-1 | 7517 |
| **0063/10** | 83876 | 8526 | 10 | 344 | 18 | 9 | 11 | 9 | 17 | ST-23 | 5–2 | 10–1 | F5-12 | 7645 |
| **0004/11** | 83864 | 23 | 10 | 5 | 18 | 9 | 11 | 9 | 17 | ST-23 | 5–2 | 10–1 | F4-1 | 2421 |
| 0014/11 | 83904 | 1625 | 10 | 5 | 18 | 14 | 11 | 9 | 17 | ST-23 | 5–1 | 2–2 | F5-8 | 2422 |
| **0102/11** | 83916 | 168 | 2 | 16 | 6 | 17 | 9 | 18 | 8 | ST-167 | 5–1 | 10–4 | F4-1 | 7517 |
| **0089/12** | 83880 | 168 | 2 | 16 | 6 | 17 | 9 | 18 | 8 | ST-167 | 5–1 | 10–4 | F4-1 | 7517 |
| **0032/13** | 83868 | 23 | 10 | 5 | 18 | 9 | 11 | 9 | 17 | ST-23 | 5–2 | 10–1 | F4-1 | 2421 |
| **0042/13** | 36323 | 23 | 10 | 5 | 18 | 9 | 11 | 9 | 17 | ST-23 | 5–2 | 10–1 | F5-12 | 89807 |
| **0060/14** | 83875 | 11017 | 8 | 5 | 6 | 17 | 8 | 18 | 8 | ST-103 | 7–2 | 13–2 | F3-9 | 169886 |
| **0024/16** | 83867 | 168 | 2 | 16 | 6 | 17 | 9 | 18 | 8 | ST-167 | 5–1 | 10–4 | F4-1 | 125578 |
| **0019/17** | 83866 | 168 | 2 | 16 | 6 | 17 | 9 | 18 | 8 | ST-167 | 5–1 | 10–4 | - | 7517 |
| **0037/18** | 101296 | 92 | 3 | 7 | 4 | 37 | 8 | 18 | 21 | ST-92 | 5–1 | 10–4 | F1-5 | 163667 |
| **0061/18** | 82068 | 1466 | 6 | 5 | 173 | 13 | 5 | 24 | 17 | ST-174 | 21 | 16 | F3-7 | 80732 |

Year of isolation is indicated in the strain ID; strains isolated from IMD are indicated in bold, isolates from carriers in normal font; VR1, VR2 = PorA variable region 1 and 2; VR = FetA variable region; ST = sequence type; cc = clonal complex, UA = clonal complex unassigned; yellow highlight = newly described allele, sequence type, or ribosomal sequence type.

Czech Republic was recorded in adolescents/young adults (age 15–19), followed by older adults (65+ years of age) and in younger children (1–4 years old) (S2 Fig).

Given the type of construction of the peptide MenB vaccines, among other benefits, they have the potential to provide protection against other meningococcal serogroups. According to the MenDeVAR index, the isolates are classified into four groups in relation to both MenB vaccines. Isolates containing one or more specific antigenic variants included in MenB vaccines are highlighted in green. Isolates containing one or more antigenic variants that showed cross-reactivity in experimental studies are highlighted in orange. Isolates for which enough data have not been available on their antigenic variants are highlighted in grey, and those carrying antigenic variants that did not show cross-reactivity in experimental studies are highlighted in red (https://pubmlst.org/organisms/neisseria-spp/mendevar). In the study collection of MenY isolates, phenomenon of potential protection can be observed particularly for the Trumenba vaccine (Table 2). The MenDeVAR index classified 32 of 43 isolates (nearly 75%) in the PubMLST database as covered by cross-reactivity. Experimental data are not yet available for the remaining 11 isolates. Regarding the Bexsero vaccine one isolate carried NadA peptide variant 8, which is included in this vaccine. One isolate appeared to be covered due to cross-reactivity and one isolate was considered as uncovered by the Bexsero vaccine based on the MenDeVAR index. For the remaining 40 isolates, experimental data for at least one antigenic variant was not available, and thus they could not be categorized based on the MenDeVAR index.

Eighteen out of 35 invasive isolates belonged to clonal complex cc23, which was detected throughout the studied years, but showed no increase in any period (S3 Fig). The phylogenetic network (Fig 1) illustrates the relatedness of all study isolates of *N. meningitidis* Y from the Czech Republic. The genetic relationships between cc23 isolates are shown in more detail in a separate phylogenetic network (Fig 2).

The most common clonal complex in our study set was cc23, which is represented by eighteen invasive and three carriage isolates. Isolates of cc23 form a separate lineage, distinct from all other isolates of *N. meningitidis* Y. The cc23 lineage splits into two sister sublineages, ST-23 and ST-569/1625. Eight of 10 ST-23 isolates form a highly compact cluster of sublineage ST-23. Sequence type ST-8526 (63/10) differs from ST-23 in the MLST genes by allele 344 of the *adk* gene, which was newly described in isolate 63/10 as well as ST-8526 itself. The other sublineage is formed by two genetically related clusters, ST-569 and ST-1625, which include almost exclusively invasive isolates of the respective sequence types. Interestingly, invasive isolate 55/02—Y: P1.7–2,13–2: F4-17: ST-23 (cc23), despite its standard MLST gene alleles, is the only cc23 outlier from sublineages ST-23 and ST-569/1625 (Fig 1). A more detailed phylogenetic analysis of cc23 isolates (Fig 2) showed that it has a unique genetic profile and is unrelated to any of the major sublineages of clonal complex cc23. In this isolate, a new ribosomal sequence type, rST-137520, and a combination of MenB vaccine antigen genes, BAST-2988, have also been described.

The second leading clonal complex of the study isolates of *N. meningitidis* Y was cc92 (Fig 1). It is represented by three invasive and four carriage isolates and is quite heterogeneous in the phylogenetic network. One lineage of clonal complex cc92 is formed by two almost identical carriage isolates, 323/93 and 338/93, along with a more distant carriage isolate 59/93 (ST-92). Another lineage of cc92, groups two genetically closely related invasive isolates 200/94 and 449/95. A more distant relatedness to this lineage was evidenced for a recent invasive isolate, 37/18. The list of isolates of clonal complex cc92 ends with an outlier carriage isolate, 224/93 which is the only one to be assigned to a different sequence type, ST-94.

Six highly related invasive isolates form lineage cc167 in the phylogenetic network, which shows relatedness to lineage cc92. All cc167 isolates carry a single sequence type, ST-

**Table 2. Molecular characterization focused on MenB vaccine antigens of *N. meningitidis* Y isolates collected in the Czech Republic from 1993 to 2018, (n = 43).**

| Strain IDs | porA VR1 | porA VR2 | nhba allele | nhba peptide | nadA allele | nadA variant | nadA peptide | fHbp allele | fHbp peptide | fHbp variant | fHbp family | BAST type | Men DeVAR (Bexsero) | Men DeVAR (Trumenba) |
|---|---|---|---|---|---|---|---|---|---|---|---|---|---|---|
| 0059/93 | 5–1 | 10–14 | 7 | 9 | - | - | - | 687 | 101 | 2 | A | 2917 | | |
| 0111/93 | 18–7 | 9 | 15 | 24 | - | - | - | 688 | 34 | 2 | A | 1479 | | |
| 0224/93 | 5–1 | 10–4 | 7 | 9 | - | - | - | 21 | 21 | 2 | A | 549 | - | fHbp 21 |
| 0323/93 | 5–1 | 10–1 | 7 | 9 | - | - | - | 687 | 101 | 2 | A | 987 | | |
| 0338/93 | 5–1 | 10–1 | 7 | 9 | - | - | - | 687 | 101 | 2 | A | 987 | | |
| 0200/94 | 5–1 | 10–4 | 7 | 9 | - | - | - | 691 | 585 | 2 | A | 1501 | | |
| 0040/95 | 5–1 | 2–2 | 13 | 8 | - | - | - | 25 | 25 | 2 | A | 227 | | fHbp 25 |
| 0449/95 | 5–1 | 10–4 | 7 | 9 | - | - | - | 691 | 585 | 2 | A | 1501 | | |
| 0008/96 | 12–1 | 16–8 | 15 | 24 | - | - | - | 302 | 245 | 1 | B | 2987 | | |
| 0264/99 | 5–1 | 2–2 | 13 | 8 | - | - | - | 25 | 25 | 2 | A | 227 | | fHbp 25 |
| 0172/01 | 5–1 | 2–2 | 13 | 8 | - | - | - | 25 | 25 | 2 | A | 227 | | fHbp 25 |
| 0179/01 | 5–1 | 2–2 | 13 | 8 | - | - | - | 25 | 25 | 2 | A | 227 | | fHbp 25 |
| 0251/01 | 5–2 | 10–1 | 10 | 7 | - | - | - | 25 | 25 | 2 | A | 228 | | fHbp 25 |
| 0055/02 | 7–2 | 13–2 | 335 | 145 | - | - | - | 21 | 21 | 2 | A | 2988 | | fHbp 21 |
| 0068/02 | 5–1 | 2–2 | 13 | 8 | - | - | - | 25 | 25 | 2 | A | 227 | | fHbp 25 |
| 0177/02 | 5–1 | 2–2 | 13 | 8 | - | - | - | 25 | 25 | 2 | A | 227 | | fHbp 25 |
| 0066/03 | 18–12 | 10–2 | 15 | 24 | - | - | - | 302 | 245 | 1 | B | 1298 | | |
| 0157/03 | 5–2 | 10–1 | 3 | 20 | - | - | - | 16 | 16 | 2 | A | 1418 | | fHbp 16 |
| 0192/04 | 5–1 | 2–2 | 13 | 8 | - | - | - | 25 | 25 | 2 | A | 227 | | fHbp 25 |
| 0267/04 | 5–1 | 2–2 | 13 | 8 | - | - | - | 25 | 25 | 2 | A | 227 | | fHbp 25 |
| 0065/05 | 5–2 | 10–1 | 3 | 20 | - | - | - | 16 | 16 | 2 | A | 1418 | | fHbp 16 |
| 0075/05 | 5–1 | 2–2 | 13 | 8 | - | - | - | 25 | 25 | 2 | A | 227 | | fHbp 25 |
| 0094/07 | 5–2 | 10–1 | 10 | 7 | - | - | - | 25 | 25 | 2 | A | 228 | | fHbp 25 |
| 0105/07 | 18–12 | 10–2 | 15 | 24 | - | - | - | 1079 | 245 | 1 | B | 1298 | | |
| 0156/07 | 18–1 | 16–4 | 3 | 20 | - | - | - | 14 | 14 | 1 | B | 2990 | fHbp 14 | fHbp 14 |
| 0190/07 | 5–1 | 10–4 | 509 | 9 | - | - | - | 23 | 23 | 2 | A | 384 | | fHbp 23 |
| 0121/08 | 5–2 | 10–1 | 10 | 7 | - | - | - | 25 | 25 | 2 | A | 228 | | fHbp 25 |
| 0125/08 | 5–2 | 10–1 | 10 | 7 | - | - | - | 25 | 25 | 2 | A | 228 | | fHbp 25 |
| 0126/08 | 5–2 | 10–1 | 10 | 7 | - | - | - | 25 | 25 | 2 | A | 228 | | fHbp 25 |
| 0049/09 | 5–2 | 10–1 | 10 | 7 | - | - | - | 25 | 25 | 2 | A | 228 | | fHbp 25 |
| 0039/10 | 5–1 | 10–4 | 509 | 9 | - | - | - | 23 | 23 | 2 | A | 384 | | fHbp 23 |
| 0063/10 | 5–2 | 10–1 | 10 | 7 | - | - | - | 25 | 25 | 2 | A | 228 | | fHbp 25 |
| 0004/11 | 5–2 | 10–1 | 10 | 7 | - | - | - | 25 | 25 | 2 | A | 228 | | fHbp 25 |
| 0014/11 | 5–1 | 2–2 | 13 | 8 | - | - | - | 25 | 25 | 2 | A | 227 | | fHbp 25 |
| 0102/11 | 5–1 | 10–4 | 509 | 9 | - | - | - | 23 | 23 | 2 | A | 384 | | fHbp 23 |
| 0089/12 | 5–1 | 10–4 | 509 | 9 | - | - | - | 23 | 23 | 2 | A | 384 | | fHbp 23 |
| 0032/13 | 5–2 | 10–1 | 10 | 7 | - | - | - | 25 | 25 | 2 | A | 228 | | fHbp 25 |
| 0042/13 | 5–2 | 10–1 | 10 | 7 | - | - | - | 25 | 25 | 2 | A | 228 | | fHbp 25 |
| 0060/14 | 7–2 | 13–2 | 15 | 24 | 109 | 4/5 | 21 | 785 | 649 | 1 | B | 2972 | | |
| 0024/16 | 5–1 | 10–4 | 509 | 9 | - | - | - | 23 | 23 | 2 | A | 384 | | fHbp 23 |
| 0019/17 | 5–1 | 10–4 | 509 | 9 | - | - | - | 23 | 23 | 2 | A | 384 | | fHbp 23 |
| 0037/18 | 5–1 | 10–4 | 7 | 9 | - | - | - | 691 | 585 | 2 | A | 1501 | | |
| 0061/18 | 21 | 16 | 9 | 6 | 80 | 2/3 | 8 | 21 | 21 | 2 | A | 14 | nadA 8 | fHbp 21 |

Year of isolation is indicated in the strain ID; strains isolated from IMD are indicated in bold, isolates from carriers in normal font; VR1, VR2 = PorA variable region 1 and 2; yellow highlight = newly described allele or the BAST type; MenDeVAR: green, orange, grey, and red highlight = graphical representation of four groups with different functional effects in relation to both MenB vaccines.

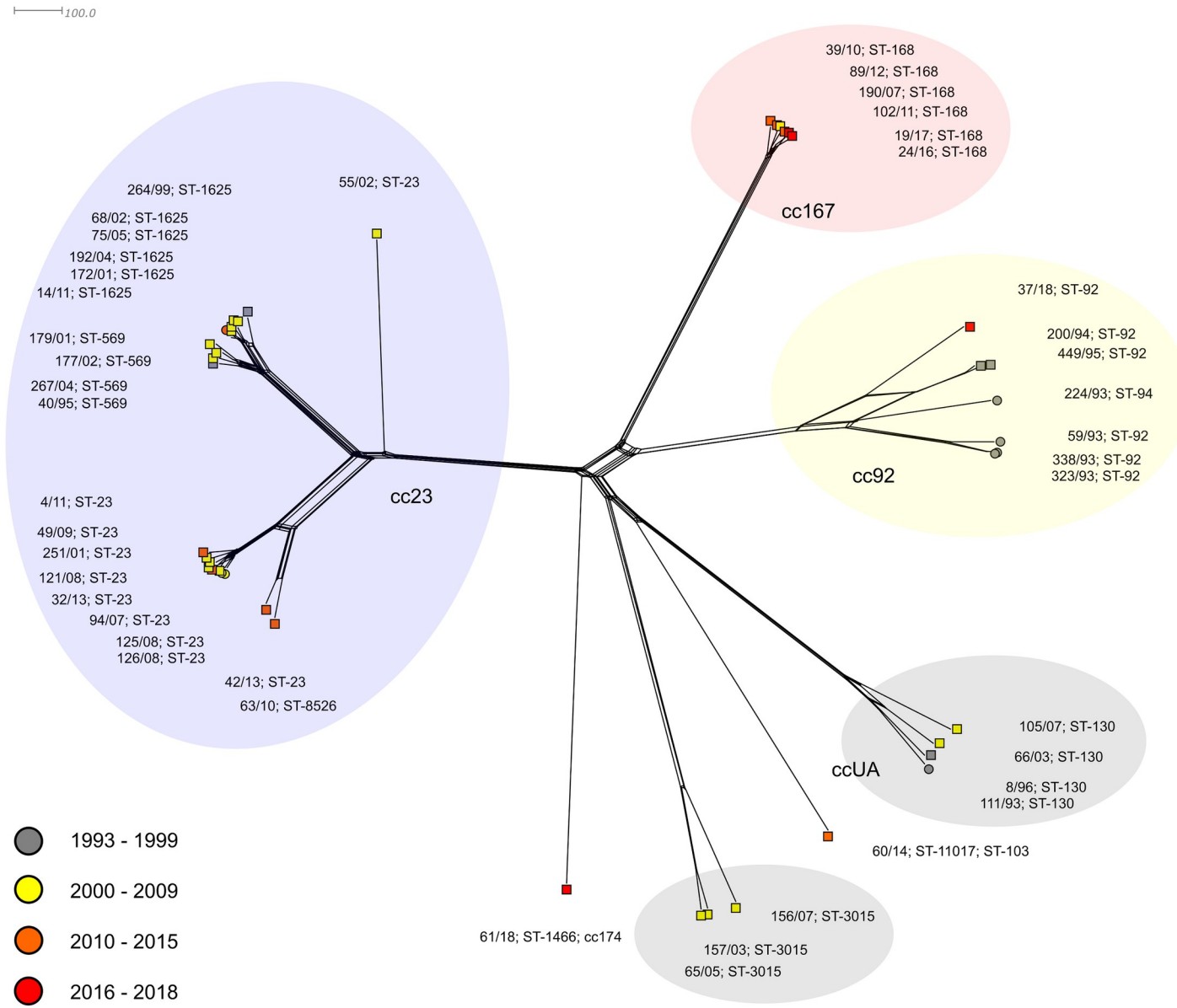

**Fig 1. Genetic relationship of *N. meningitidis* Y isolates collected in the Czech Republic from 1993 to 2018.** Neighbour-net network of MenY isolates (n = 43). Isolates are coloured according to detection year and labelled by their NRL number, ST, and cc. Invasive isolates are marked with a square, carriage isolates with a circle.

168, and BAST-384. In isolate 19/17, a single nucleotide deletion occurred in the variable region (VR) of the *fetA* gene, resulting in a reading frame shift and internal stop codon emergence. This mutated *fetA* allele, unable to produce functional peptide, was registered in the PubMLST database as allele 2816. Another comparatively heterogeneous lineage groups four ST-130 isolates. For isolate 105/07, a new allele variant of the *fHbp* gene (1079) was described, which differs from the original allele 302 by two synonymous substitutions, thus encoding the same peptide variant, 245. An outlier invasive isolate of clonal complex cc103 appears to be distantly related to the ST-130 lineage of isolates. Based on this outlier, a new sequence type, ST-11017, a new rST-169886, and a new combination of MenB vaccine

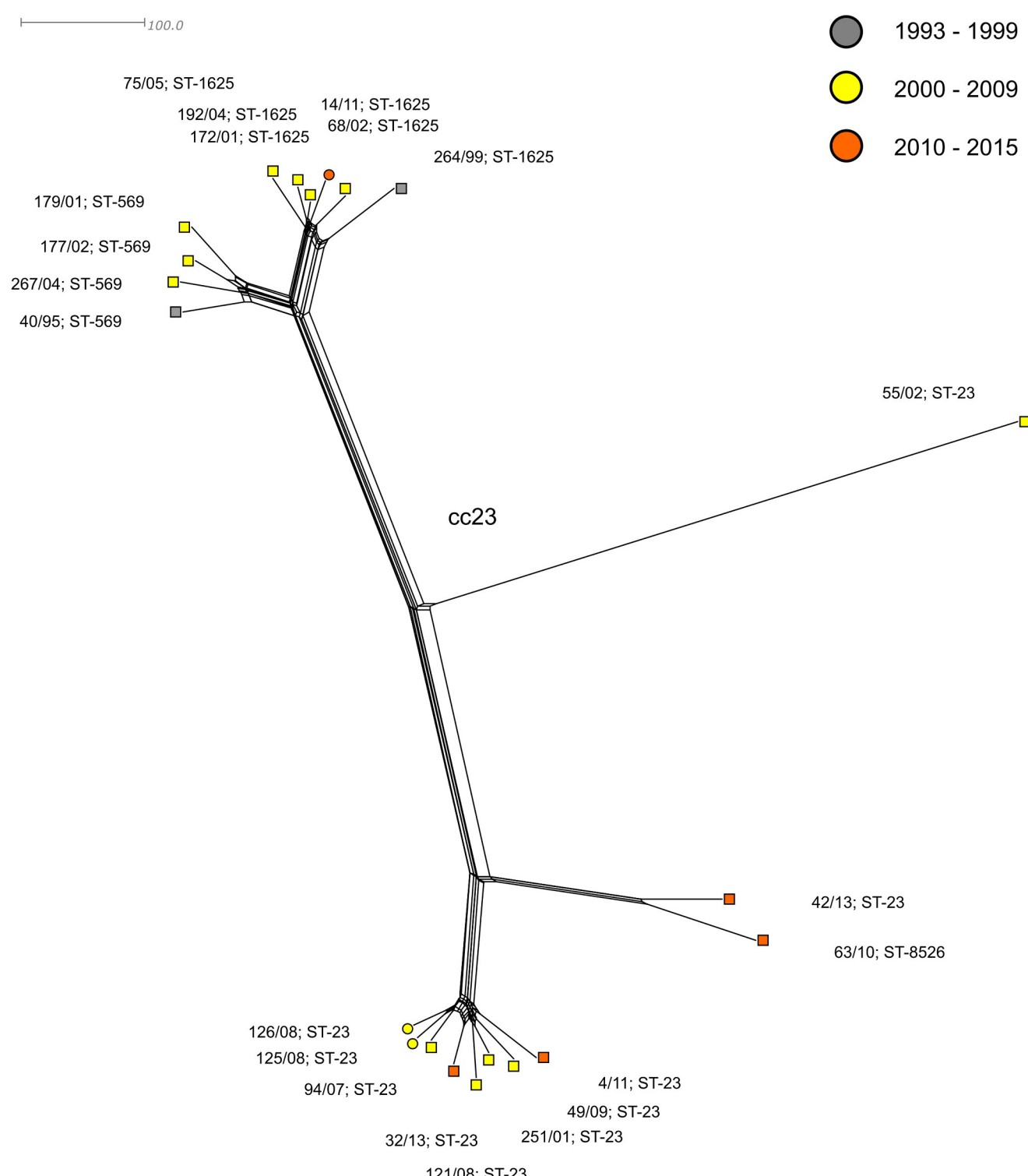

**Fig 2. Genetic relationship of *N. meningitidis* Y isolates collected in the Czech Republic from 1993 to 2018, clonal complex cc23.** Neighbour-net network of MenY isolates cc23 (n = 21). Isolates are coloured according to detection year and labelled by their NRL number, ST, and cc. Invasive isolates are marked with a square, carriage isolates with a circle.

antigen genes, BAST-2972, were defined. Even more distant relatedness to ST-130 isolates is seen in a lineage of three ST-3015 (ccUA) isolates recovered between 2003 and 2007. The last isolate from our collection– 61/18 (P1.21,16: F3-7; ST-1466 (cc174); rST-80732; BAST-14) is a phylogenetic outlier and does not show any relatedness to other isolates of *N. meningitidis* Y.

## Czech and European isolates of *N. meningitidis* Y

The phylogenetic network (Fig 3A and 3B) shows the levels of relatedness among European invasive isolates of *N. meningitidis* Y, including Czech isolates. These were isolates from the following countries: UK (n = 612), Sweden (n = 112), France (n = 62), Germany (n = 49), the Netherlands (n = 46), Czech Republic (n = 35), Ireland (n = 29), Austria (n = 12), Norway (n = 11), Italy (n = 10), Finland (n = 6), Slovenia (n = 6), Malta (n = 3), Cyprus (n = 1), and Greece (n = 1).

The general view shows that one part of the European isolates of *N. meningitidis* Y form a large subpopulation of several lineages of clonal complex cc23 in the phylogenetic network, while the other part form separate genetic lineages consistent with the respective clonal complexes.

The large cc23 subpopulation consisting of several more or less related lineages includes 18 invasive isolates of clonal complex cc23 from the Czech Republic. All cc23 lineages are represented by isolates from all time periods, and none of them can be considered as historical or modern (Fig 3A). The dominance of isolates from the years 2010–2020 is caused by the overall low proportion of isolates from the years 1986–2009 in the phylogenetic analysis. An identical phylogenetic network that shows geographical origin of isolates in colour provides a completely different view (Fig 3B). A distant lineage (I), grouping mostly ST-23 isolates from the UK can be noticed. In three similar and genetically closely related lineages (II, III, and IV), the UK isolates are dominant again; however, isolates from neighbouring countries are also significantly represented. Most isolates from these three lineages belong to sequence type ST-1655. Another cc23 lineage comprises mostly Swedish ST-23 isolates (V). Close to the Swedish lineage is a geographically diverse lineage (VI) of isolates from continental Europe where ST-23 and sequence types derived from it are dominant. Apart from a cluster of six Czech isolates (251/01, 94/07, 121/08, 49/09, 4/11, and 32/13), this lineage includes primarily isolates from Germany, Austria, the Netherlands, and Slovenia. Czech isolates 63/10 and 42/13, which do not cluster together, belong to a genetically more distant lineage comprising mostly ST-23 isolates from France (VII). The last Czech cc23 isolate is 55/02 (ST-23), which, like the previous Czech phylogenetic network, is an outlier, and from its position it is evident that it is highly genetically distant from all other European cc23 isolates.

European isolates of *N. meningitidis* Y not belonging to cc23 show no significant trend in temporal distribution. The phylogenetic network which defines these isolates as genetically highly distant from cc23 subpopulation splits into three major lineages. The first one is a compact lineage, cc174, consisting exclusively of ST-1466 isolates including isolate 61/18 from the Czech Republic. The second lineage is highly heterogeneous, both geographically and genetically. It groups isolates ccUA, cc22, cc103, cc11, and cc865. The third major lineage in terms of heterogeneity appears in between the two former lineages. At the bottom, a sublineage of five cc92 isolates separates from it, formed by a cluster of three Czech isolates ST-92 along with a UK and Dutch isolates (ST-784). Other members of this third major lineage are almost exclusively cc167 isolates, including the cluster of six ST-168 isolates from the Czech Republic.

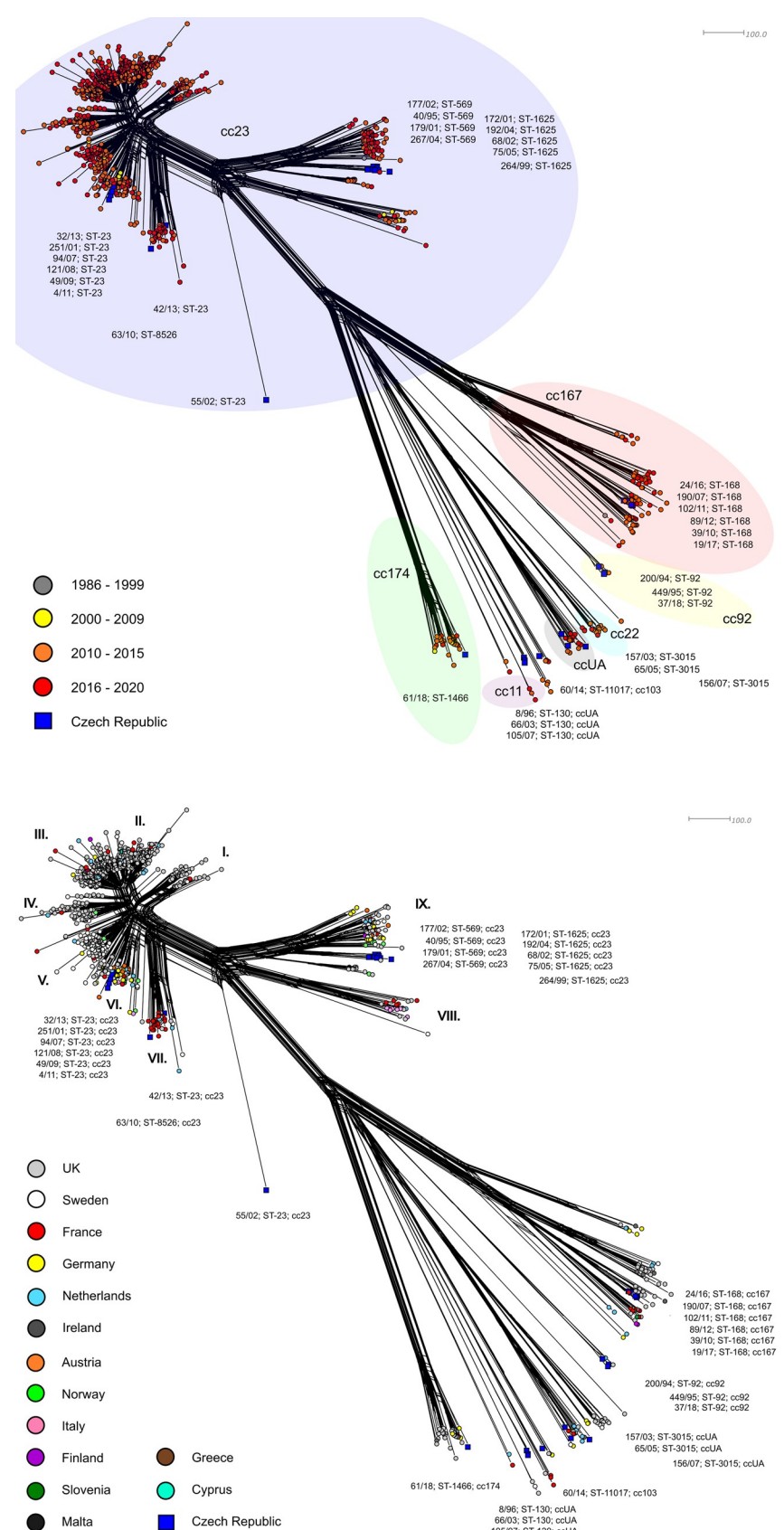

**Fig 3. Genetic diversity of *N. meningitidis* Y invasive isolates from the Czech Republic and European countries.**
Isolates (n = 995) were collected from 1986 to 2020. Invasive isolates from Czech Republic (n = 35) are marked with a
dark blue square and labelled by their NRL number, ST, and cc. A) Isolates are coloured according to detection year:
1986–1999 (n = 8), 2000–2009 (n = 28), 2010–2015 (n = 524), 2016–2020 (n = 435). B) Isolates are coloured according
to the country of origin. The individual lineages of the cc23 clonal complex are marked by Roman numerals (I—IX).

## Czech and non-European isolates of *N. meningitidis* Y

The last phylogenetic network (Fig 4A and 4B) compares invasive isolates of *N. meningitidis* Y
from the Czech Republic and non-European countries. In this selection, the MenY isolates
from the USA are dominant (n = 315), followed by Japan (n = 24), New Zealand (n = 13),
Mexico (n = 8), Canada (n = 4), Turkey (n = 3), China (n = 2), Morocco (n = 2), Burkina Faso
(n = 2), Chile (n = 1), and Bangladesh (n = 1). Along with 35 isolates from the Czech Republic,
the study set comprises 410 invasive isolates.

A general view of the phylogenetic network shows again a large subpopulation of cc23 iso-
lates while isolates of *N. meningitidis* Y of other clonal complexes form several separate more
or less numerous lineages on the opposite side.

Unlike the European cc23 isolates, non-European cc23 isolates appear less homogeneous.
They form four separate, genetically distinct lineages. Furthermore, they differ in temporal dis-
tribution of isolates in individual lineages (Fig 4A). The first cc23 lineage (I) comprising exclu-
sively of isolates from the USA and Mexico is represented mostly by isolates ST-3587
recovered after 2016 (Fig 4B). Another cc23 lineage (II) groups two main sister sublineages of
isolates from the USA, with ST-23 and related ST-3532 being dominant. In addition to the two
main sublineages, three other sublineages separate from the lineage II of cc23 isolates. One of
these sublineages groups two clusters of 9 isolates from the Czech Republic (ST-569 and ST-
1625) along with three isolates from the USA. Unlike the first major cc23 lineage, the second
one is evenly represented by isolates from all time periods. The third cc23 lineage is geographi-
cally more diverse (III) and mostly assigned to sequence type ST-23. Two sublineages diverge
from this lineage: one with dominant ST-23 groups isolates from the USA along with two
Czech isolates and sublineage ST-1655 comprising isolates from the USA, Japan, New Zealand,
and China. The remaining six Czech ST-23 isolates cluster together within the major lineage.
The last, fourth cc23 lineage (IV) only includes two isolates, 55/02 (ST-23) from the Czech
Republic collected in 2002, which so far occupied a completely separate position in the phylo-
genetic networks, along with a related American isolate, M28856 (ST-6519) from 2014.

Non-European isolates of *N. meningitidis* Y which do not belong to clonal complex cc23
again form several lineages genetically distant from each other on the phylogenetic network.
In comparison with the European isolates, non-European isolates are less represented by other
clonal complexes, as e.g., two genetic lineages are only formed by isolates from the Czech
Republic (ST-92 and ST-130). The only exception is the richly represented lineage cc167, con-
sisting mainly of American isolates ST-1624. Another lineage is represented exclusively by ST-
1466 (cc174) isolates, including isolate 61/18 from the Czech Republic.

## Discussion

We analysed WGS data of all *N. meningitidis* Y isolates received from IMD in the Czech
Republic in the studied period. WGS showed a higher discrimination power and provided
more accurate data on molecular characteristics and genetic relationships among invasive
*N. meningitidis* isolates [18]. However, it is costly for the NRL to perform genomic surveil-
lance for all *N. meningitidis* isolated from IMD and therefore we focus on serogroups that
cause a high case fatality rate of this disease. Our study was focused on the analysis of WGS

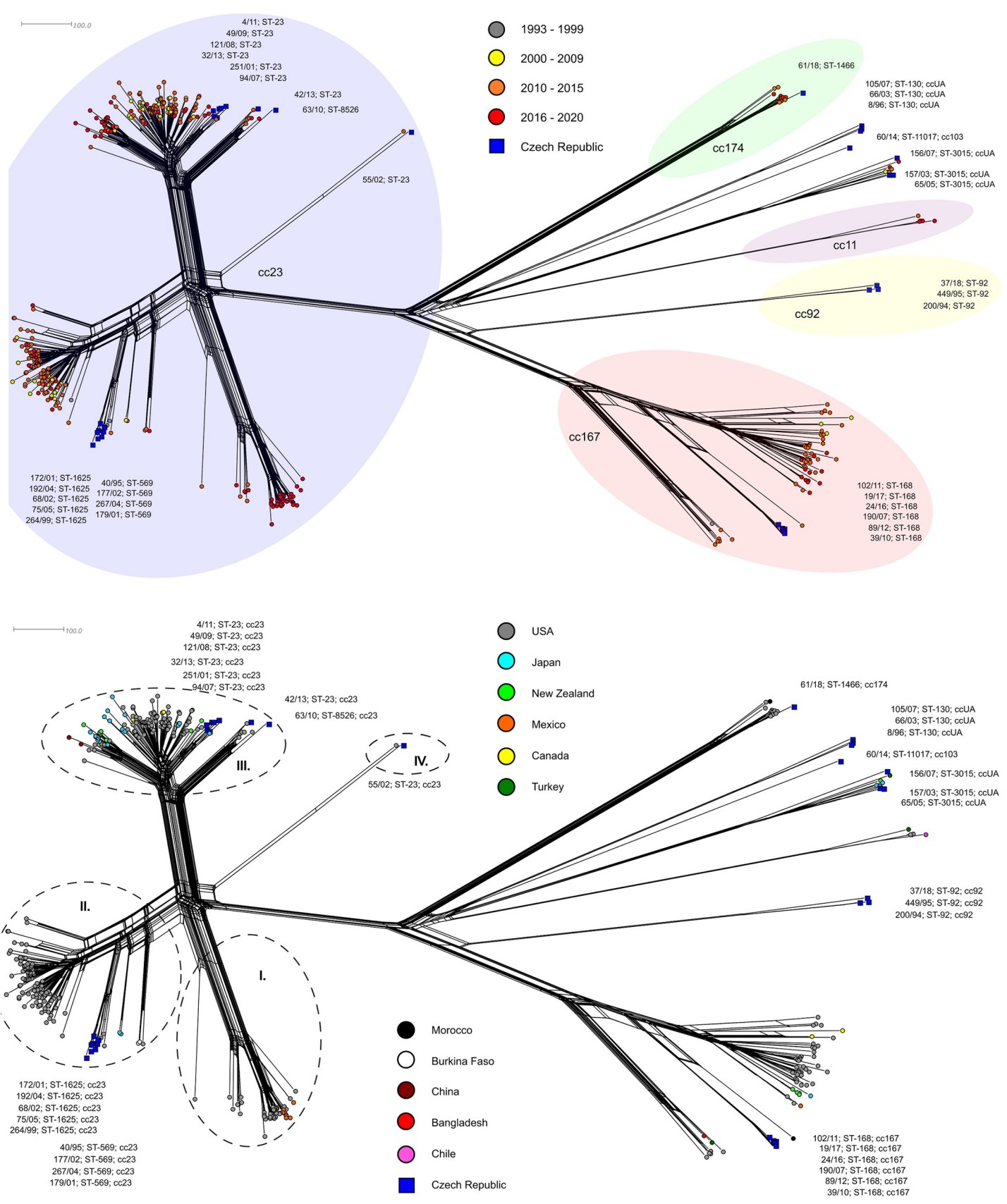

**Fig 4. Genetic diversity of *N. meningitidis* Y invasive isolates from the Czech Republic and non-European countries.** Isolates (n = 410) were collected from 1993 to 2020. Invasive isolates from Czech Republic (n = 35) are marked with a dark blue square and labelled by their NRL number, ST, and cc. A) Isolates are coloured according to detection year: 1993–1999 (n = 14), 2000–2009 (n = 53), 2010–2015 (n = 213), 2016–2020 (n = 130). B) Isolates are coloured according to the country of origin. The individual lineages of the cc23 clonal complex are marked by Roman numerals (I–IV).

data of Czech isolates of *N. meningitidis* serogroup Y and their comparison with foreign data because this serogroup causes high IMD mortality worldwide and its increase has recently been recorded [6–14]. Comparison of WGS data of Czech isolates of *N. meningitidis* Y with foreign ones was also performed because in a previous study we found that the population of Czech isolates of *N. meningitidis* serogroup W was genetically different from European ones [17].

A limitation of our study is a low number of *N. meningitidis* Y isolates. However, this corresponds to the real epidemiological situation in the Czech Republic, i.e. low frequency of IMD caused by serogroup Y, although its rise has been recorded recently.

The European invasive isolates of *N. meningitidis* Y, including Czech isolates, can be considered as a relatively heterogeneous population. Although about two thirds of which are genetically closely related and belong to cc23, the remaining isolates are assigned to many other clonal complexes and show a very low relatedness to cc23 isolates and to each other. Isolates of *N. meningitidis* Y belonging to cc23 and other clonal complexes form sublineages within genetic lineages, defined mainly geographically. The population of invasive isolates of *N. meningitidis* Y appears to be constant over time compared to non-European isolates. The Czech isolates of *N. meningitidis* Y follow the trend observed for European isolates. They belong to several different clonal complexes and mostly cluster together within individual genetic lineages.

The non-European population of isolates of *N. meningitidis* Y appears to be less diverse in terms of clonal complexes as compared to the European one. Most isolates belong to complexes cc23 and cc167. A higher genetic diversity can be seen within individual clonal complexes–particularly in subpopulation cc23. The non-European population of *N. meningitidis* Y also shows a different time distribution of isolates. Within clonal complexes, several modern sublineages appear, containing mostly isolates collected after 2016. The Czech isolates of *N. meningitidis* Y, added to the non-European isolates for comparison, do not follow this non-European trend. Their clonal heterogeneity considerably contributes to the overall heterogeneity of the phylogenetic network, which contains Czech and non-European isolates.

The worldwide increase in IMD caused by serogroup Y has led to increased efforts to study *N. meningitidis* Y by the latest molecular methods. The analysis of WGS data on Swedish IMD isolates of *N. meningitidis* Y from 1995–2012 revealed two major branches of clonal complex cc23, one of which corresponded to YI strain type (ST-23) and the other to YII and YIII strain types [14]. The comparison of the Swedish WGS data with those from England, Wales, and Northern Ireland confirmed the spread of these two branches across continents over the last few years, although these branches were referred to differently in studies. IMD isolates of serogroup Y collected in England and Wales in 2007–2009 belonged to four major clonal complexes: cc23 (56%), cc174 (21%), cc167 (11%), and cc22 (8%). The rise in IMD caused by serogroup Y in 2009 was mainly due to ST-1655 (cc23) [10]. The variability of *N. meningitidis* Y cc23 was also reported for Italian IMD isolates from 2007–2013, assigned to nine different STs (ST-23, ST-10098, ST-1665, ST-10348, ST-9326, ST-2533, ST-9253, ST-3171, and ST-2692), with a predominance of ST-23 [13]. In our study, ST-23 was also prevalent among isolates of *N. meningitidis* Y cc23, but other STs (ST-1625, ST-569, and ST-8526) differed from those identified in the Italian study. Isolates ST-1655 (cc23) that caused the rise in IMD in England and Wales were not observed in our collection. IMD

isolates of serogroup Y collected in Canada in 1999–2003 were assigned to two major clonal complexes: cc23 and cc167 [37]. The prevalence of clonal complexes cc23 and cc167 found in our study is consistent with the Canadian data, as is the prevalence of ST-23 in clonal complex cc23. The comparison of IMD isolates of serogroup Y from the United States, South Africa, and Israel collected in 1999–2002 showed the prevalence of two clonal complexes, cc23 and cc175. However, differences were found in the representation of clonal complexes between countries: the USA and Israel reported the prevalence of cc23, while in South Africa, the prevalence of cc175 was observed [38]. There was not a single cc175 isolate in our collection.

In conclusion, the WGS analysis showed the population of Czech *N. meningitidis* Y isolates as relatively heterogeneous, containing a large number of genetic lineages. The most represented clonal complex in Czech *N. meningitidis* Y isolates was cc23, which forms a separate lineage, distinct from all other isolates of *N. meningitidis* Y. Isolates of other clonal complexes show a very low relatedness to cc23 isolates and to each other. The comparison with foreign WGS data showed that within the main genetic lineages, which are defined by clonal complexes, Czech isolates of *N. meningitidis* Y, similar to European ones, mostly cluster together and form geographical sublineages. The Czech isolates of *N. meningitidis* Y generally follow the trend observed for European isolates in contrast to Czech isolates of serogroup W, for which a different trend was observed in our previous publication [17]. This result was one of the bases for updating the recommended vaccination strategy and in the Czech Republic the conjugated tetravaccine ACYW started to be used instead of the previous conjugated monovaccine C. In the future, we would like to prepare similar population studies with isolates of serogroups B and C, which are constantly dominant in the Czech Republic.

## Supporting information

**S1 Fig. Serogroups of *N. meningitidis* causing invasive meningococcal disease, Czech Republic, 1993–2018, surveillance data.**
(PDF)

**S2 Fig. Clonal complexes of *N. meningitidis* Y causing invasive meningococcal disease, Czech Republic, 1993–2018, age groups, surveillance data.**
(PDF)

**S3 Fig. Clonal complexes of *N. meningitidis* Y causing invasive meningococcal disease, Czech Republic, 1993–2018, surveillance data.**
(PDF)

## Acknowledgments

This publication made use of the PubMLST website (https://pubmlst.org/) developed by Keith Jolley (Jolley & Maiden 2010, BMC Bioinformatics, 11:595) and sited at the University of Oxford. We thank Keith Jolley from the University of Oxford for editing the text and language proofing.

## Author Contributions

**Conceptualization:** Michal Honskus, Pavla Krizova.

**Data curation:** Michal Honskus, Zuzana Okonji, Martin Musilek.

**Formal analysis:** Michal Honskus, Pavla Krizova.

**Funding acquisition:** Pavla Krizova.

**Investigation:** Michal Honskus, Zuzana Okonji, Martin Musilek, Pavla Krizova.

**Methodology:** Michal Honskus.

**Project administration:** Pavla Krizova.

**Resources:** Pavla Krizova.

**Software:** Michal Honskus.

**Supervision:** Pavla Krizova.

**Validation:** Michal Honskus, Pavla Krizova.

**Visualization:** Michal Honskus, Pavla Krizova.

**Writing – original draft:** Michal Honskus, Pavla Krizova.

**Writing – review & editing:** Michal Honskus, Zuzana Okonji, Martin Musilek, Pavla Krizova.

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
