## [Decision Letter · Decision Letter 0]

29 Oct 2021

PONE-D-21-29616Whole Genome Sequencing of Neisseria meningitidis Y isolates collected in the Czech Republic in 1993-2018PLOS ONE

Dear Dr. Krizova,

Thank you for submitting your manuscript to PLOS ONE. After careful consideration, we feel that it has merit but does not fully meet PLOS ONE’s publication criteria as it currently stands. Therefore, we invite you to submit a revised version of the manuscript that addresses the points raised during the review process.

I have received the reviews of your manuscript. While your paper addresses an interesting question and provide valuable information of the epidemiologically situation of *Neisseria meningitides*, there are significant concern both about the presentation as well as the readability of the manuscript.  In particular, the rationale of the study needs to be strengthen and the aim of the study need to be explained in the abstract and introduction.  The writing was inadequate and accordingly, the data were not following a clear logical line of thought. In addition there were numerous issues identified where additional experimentation and documentation is needed, please see reviewers’ insightful comments below.  On a more personal level, I also have a few issues with the manuscript that needs to be addressed, please see specific comments below.  Furthermore, the quality of the language needs to be improved.  Please have a fluent, preferably native, English-language speaker thoroughly copyedit your manuscript for language usage, spelling, and grammar.  If you do not know anyone who can do this, we suggest you use a professional language editing or copyediting service. 

Specific comments:

Line 49 – 56, combine these two paragraphs into one.Line 158, please specify which ribosomal gene?Expand the explanation of Table 1 further. For examples, Most prevalent is cc23 with 21 isolates (ST-23, 569,1625, and 8526).  Followed by cc92 (ST-92 and ST-94) with 7 isolates. and cc167 (ST-168) with 6 isolates, as well as one isolate each for cc103 (ST11017) and cc174 (ST1466), etc.Table 1 needs substantial editing to provide more relevant information. A few suggestion here:  1. Year of isolation is indicated in the strain ID? Is it necessary to have separate year of isolation? 2. Symptoms, you can probably delete this column and put * on the strains ID isolated from carrier and explained in the note; 3) Delete columns of serogroup and genogroup since all the isolates belong to the same serogroup and genogroup; 4) Add column to indicate clonal complex, BAST, and rST Note for Table 1 needs significant editing to provide relevant information:  1) Unless the authors added column indicating cc, the explanation for cc and UA is unnecessary; 2) Line 183 – 185, not sure that these are relevant to the current Table 1.Line 187 – 193, combine these two paragraphs into one.Line 307 – 310, are these sentence referring to one of the figures?Line 314, please explain Trumenba.Line 317 – 322, These are confusing sentences. Not sure what the authors wish to convey?Line 356 – 357, awkward sentence, please rephrase for clarity.Discussion needs to focus on discussing the results first and how it compared to other literatures.

We look forward to receiving your revised manuscript.

Kind regards,

Baochuan Lin, Ph.D.

Academic Editor

PLOS ONE

Journal Requirements:

Reviewers' comments:

Reviewer's Responses to Questions

**Comments to the Author**

1. Is the manuscript technically sound, and do the data support the conclusions?

Reviewer #1: Yes

Reviewer #2: Yes

Reviewer #3: Partly

2. Has the statistical analysis been performed appropriately and rigorously? 

Reviewer #1: N/A

Reviewer #2: N/A

Reviewer #3: N/A

3. Have the authors made all data underlying the findings in their manuscript fully available?

Reviewer #1: Yes

Reviewer #2: Yes

Reviewer #3: Yes

4. Is the manuscript presented in an intelligible fashion and written in standard English?

Reviewer #1: Yes

Reviewer #2: No

Reviewer #3: No

5. Review Comments to the Author

Reviewer #1: 1- line 33 -37 please rephrase it so it will be more clear

2-In the introduction we have to highlight the importance of our research why did you do sequencing for serogroup Y and not the main groups ,if there were increase in serogroup Y after introducing the MenC and MenB ,now as you mentioned there are 2 quadrivalent vaccines covering the Y serogroup,what is the outcome after doing WGS and the relation to the current vaccine.3-please try to summarize the description of each figure without going for much details. 4-in the discussion we have to mention our results and compare it with different countries

Reviewer #2: The manuscript entitled “Whole Genome Sequencing of Neisseria meningitidis Y isolates collected in the Czech Republic in 1993-2018” compares meningococcal serogroup Y genomes from the Czech Republic with other European and non-European genomes in order to relate the lineages in the Czech Republic to a global setting. Vaccination against serogroups ACWY and B is recommended in the Czech Republic and the study also includes characterization on theoretical vaccine coverage. Although the results from this study is an important contribution to the long-term epidemiological surveillance of meningococcal isolates in general, and the emergent serogroup Y in particular, the manuscript needs to be clarified and restructured and especially the results and discussion need a major overview according to the points suggested below.

Major points:

1. The aim of the study is not entirely clear to me and some analyses in this study are incompletely incorporated into the narrative. For example, the theoretical vaccine coverage BAST and MenDeVAR index is not described in M&M and it is not discussed in the discussion section what this information adds. Additionally, although very important, without a clearer aim it is unclear why the Czech isolates were compared to European isolates and then non-European isolates.

2. The results section lacks a clear narrative and is difficult to get through. Although there is no limit on word count, the manuscript is 6000 words long and seems to be unbalanced, 3 pages of introduction, 13 pages of results, 3 pages of discussion. The results section would benefit from being shortened and some parts moved to the discussion.

3. Figure S1 gives the reader an important sense of how the serogroup Y incidence, although increasing, is still very low in the Czech Republic compared to serogroups B and C. The authors could argue that it is rather the high mortality rate of serogroup Y that is the issue but this needs to be clearer as the introduction focuses on the increased serogroup Y IMD incidence.

4. Everything is relative but calling the serogroup Y isolates heterogeneous when 2/3 belonged to cc23 and then looking at non-European isolates which were even more heterogeneous, maybe then one would say that the Czech isolates were quite homogeneous? This could be re-phrased to be a more “soft” statement throughout the manuscript.

5. The discussion is incomplete and mostly mentions other studies. The most important findings should be highlighted and how the results adds to the field of research, possible limitations with the study and future directions should be added.

6. The manuscript would benefit from a general language revision, especially in regard to grammatical tense.

Minor points:

All figures would benefit from having the N= written in the figure legends instead of in the main text. Abbreviations NRL, ST and cc in the legends should be explained.

Line 20-22: The material and methods section of the abstract is somewhat short, it does not for example mention that the Czech isolates were compared to other countries.

Line 59: The cause of shifting meningococcal serogroup incidences in meningococci is not completely known and vaccination programmes is probably only part of the explanation, the sentence could be rephrased to e.g. “…partly due to vaccination programmes”.

Line 73: Please add a reference to this statement.

Line 78: The study was from 2015, not 2014.

Line 80: WGS was done in reference number 14.

Line 88: What is the timeline for the numbers on serogroup distribution, since 1993?

Introduction: please add numbers on IMD incidence in the Czech Republic. These would be interesting to see in conjunction to the vaccine strategy.

Line 128: There is no information about the healthy carriers, in which setting were these sampled? What was the purpose of these samples in the study?

Material and methods: The BAST, MenDeVAR index and rMLST are not mentioned.

Table 1: The entire table is not visible. The abbreviations “CAR” and “PAC” are not explained.

Line 215-216: Please give an estimation on how many the minimal genetic differences are.

Line 295-303: This should be mentioned in the Material and Methods section.

Line 479: Please use “n=3” instead of “3x ST-2880”, should be changed throughout the manuscript.

Line 494-503: This type of reasoning is better fitted for the discussion section rather than the results.

Line 515: The discussion starts with comparisons of which STs dominates in different countries, suggestion to add that the YI strain in Sweden belongs to ST-23.

Discussion: In the introduction it is stated that molecular characterization of serogroup Y provides background data for vaccination strategy, how the results from the current study will impact vaccination strategies could be discussed.

Fig 3-4: The roman letters are not explained in the figure legend.

Reviewer #3: Dear author,

Your work represents a nice work and adds valuable information of the epidemiologically situation based on whole genome sequencing of Neisseria meningitides serogroup Y isolates, collected in the Czech Republic during 1993-2018, and also its relation to other international isolates. The manuscript though needs extensive revisions, especially regarding the structure and focus of the scope of aim, i.e all parts regarding vaccination need to be shortened or in some places even taken away.

You can see more in detail below.

1. A specified aim is missing both in the manuscript and in the abstract.

2. The introduction is too long and need to be both structured and written in a logic way. For example, the section about vaccination is to detailed and needs to be shortened substantially, it should also be mentioned that the incidence and variation of causative serogroups varies over time not only due to vaccination but also natural fluctuations.

3. In M&M the collection of carrier isolates need to be described how it was done.

4. All subheadings (in the entire manuscript) need to be optimised and shortened, i.e “DNA extraction, WGS protocol, and WGS data analysis and visualization” could be Whole genome sequencing and analysis or even divided into sections.

5. The technically sound of the sequencing must be improved.

6. All abbreviations in the Tables and figures need to be explained in footnotes or used without abbreviation and print full name, especially in the headings (should be possible to read stand-alone).

7. The result section is too long and should only include the actual results and not new background information or interpretations. Do not repeat all results that can be found in tables and figures, just the most important findings.

8. The interpretations can be moved to the section of discussion. Also do not include all results from every country in the discussion only the comparison.

9. The conclusion (both in manuscript and abstract) needs to answer an aim (that first has to be defined…).

10. All figures need better solution.

6. PLOS authors have the option to publish the peer review history of their article (what does this mean?). If published, this will include your full peer review and any attached files.

Reviewer #1: **Yes: **Manal Hamed

Reviewer #2: No

Reviewer #3: No

---

## [Author Response · Author response to Decision Letter 0]

7 Dec 2021

PONE-D-21-29616

Whole Genome Sequencing of Neisseria meningitidis Y isolates collected in the Czech Republic in 1993-2018

PLOS ONE

The authors thank the academic editor and three reviewers for their valuable comments and recommendations. Most of them were accepted - see the answers to the points below.

Response to the academic editor:

While your paper addresses an interesting question and provide valuable information of the epidemiologically situation of Neisseria meningitides, there are significant concern both about the presentation as well as the readability of the manuscript. In particular, the rationale of the study needs to be strengthen and the aim of the study need to be explained in the abstract and introduction. The writing was inadequate and accordingly, the data were not following a clear logical line of thought. In addition there were numerous issues identified where additional experimentation and documentation is needed, please see reviewers’ insightful comments below. On a more personal level, I also have a few issues with the manuscript that needs to be addressed, please see specific comments below. Furthermore, the quality of the language needs to be improved. Please have a fluent, preferably native, English-language speaker thoroughly copyedit your manuscript for language usage, spelling, and grammar. If you do not know anyone who can do this, we suggest you use a professional language editing or copyediting service. 

These recommendations were mostly accepted – please see in the enclosure „Revised Manuscript with Track Changes“. 

Specific comments:

1. Line 49 – 56, combine these two paragraphs into one.

This recommendation was accepted – please see in the enclosure „Revised Manuscript with Track Changes“.

2 Line 158, please specify which ribosomal gene?

This recommendation was accepted – please see in the enclosure „Revised Manuscript with Track Changes“.

3. Expand the explanation of Table 1 further. For examples, Most prevalent is cc23 with 21 isolates (ST-23, 569,1625, and 8526). Followed by cc92 (ST-92 and ST-94) with 7 isolates. and cc167 (ST-168) with 6 isolates, as well as one isolate each for cc103 (ST11017) and cc174 (ST1466), etc.

This recommendation was accepted – please see in the enclosure „Revised Manuscript with Track Changes“.

4. Table 1 needs substantial editing to provide more relevant information. A few suggestion here: 

1) Year of isolation is indicated in the strain ID? Is it necessary to have separate year of isolation? 

This recommendation was accepted – please see in the enclosure „Revised Manuscript with Track Changes“. 2) Symptoms, you can probably delete this column and put * on the strains ID isolated from carrier and explained in the note; This recommendation was accepted – please see in the enclosure „Revised Manuscript with Track Changes“. 3) Delete columns of serogroup and genogroup since all the isolates belong to the same serogroup and genogroup; This recommendation was accepted – please see in the enclosure „Revised Manuscript with Track Changes“.

4) Add column to indicate clonal complex, BAST, and rST The columns indicating clonal complex, BAST and rST were already in Table 1 that was in the first version of the submitted manuscript. Unfortunately, they were not visible. In the revised version of the manuscript which we are now submitting, Table 1 was divided into Table 1 and Table 2 and all columns are visible.

5. Note for Table 1 needs significant editing to provide relevant information: 1) Unless the authors added column indicating cc, the explanation for cc and UA is unnecessary; As explained above, unfortunately not all Table 1 columns were visible in the first version of the manuscript. In the revised version of the manuscript which we are now submitting, Table 1 was divided into Table 1 and Table 2 and all columns are visible. The explanation for cc and UA is necessary for Table 1. 2) Line 183 – 185, not sure that these are relevant to the current Table 1. As explained above, unfortunately not all Table 1 columns were visible in the first version of the manuscript. In the revised version of the manuscript which we are now submitting, Table 1 was divided into Table 1 and Table 2 and all columns are visible. This text is relevant for Table 1 and Table 2.

6. Line 187 – 193, combine these two paragraphs into one.

This recommendation was accepted – please see in the enclosure „Revised Manuscript with Track Changes“.

7. Line 307 – 310, are these sentence referring to one of the figures?

As explained above, unfortunately not all Table 1 columns were visible in the first version of the manuscript. In the revised version of the manuscript which we are now submitting, Table 1 was divided into Table 1 and Table 2 and all columns are visible. This text is relevant for Table 2.

8. Line 314, please explain Trumenba.

This recommendation was accepted – please see in the enclosure „Revised Manuscript with Track Changes“.

9. Line 317 – 322, These are confusing sentences. Not sure what the authors wish to convey?

As explained above, unfortunately not all Table 1 columns were visible in the first version of the manuscript. In the revised version of the manuscript which we are now submitting, Table 1 was divided into Table 1 and Table 2 and all columns are visible. This text is relevant for Table 2.

10. Line 356 – 357, awkward sentence, please rephrase for clarity.

This recommendation was accepted – please see in the enclosure „Revised Manuscript with Track Changes“.

11. Discussion needs to focus on discussing the results first and how it compared to other literatures.

This recommendation was accepted – please see in the enclosure „Revised Manuscript with Track Changes“.

Response to the Reviewer #1:

1- line 33 -37 please rephrase it so it will be more clear

2-In the introduction we have to highlight the importance of our research why did you do sequencing for serogroup Y and not the main groups ,if there were increase in serogroup Y after introducing the MenC and MenB ,now as you mentioned there are 2 quadrivalent vaccines covering the Y serogroup,what is the outcome after doing WGS and the relation to the current vaccine. 3-please try to summarize the description of each figure without going for much details. 4-in the discussion we have to mention our results and compare it with different countries

These recommendations were accepted – please see in the enclosure „Revised Manuscript with Track Changes“.

Response to the Reviewer #2: 

The manuscript entitled “Whole Genome Sequencing of Neisseria meningitidis Y isolates collected in the Czech Republic in 1993-2018” compares meningococcal serogroup Y genomes from the Czech Republic with other European and non-European genomes in order to relate the lineages in the Czech Republic to a global setting. Vaccination against serogroups ACWY and B is recommended in the Czech Republic and the study also includes characterization on theoretical vaccine coverage. Although the results from this study is an important contribution to the long-term epidemiological surveillance of meningococcal isolates in general, and the emergent serogroup Y in particular, the manuscript needs to be clarified and restructured and especially the results and discussion need a major overview according to the points suggested below.

These recommendations were accepted – please see in the enclosure „Revised Manuscript with Track Changes“.

Major points:

1. The aim of the study is not entirely clear to me and some analyses in this study are incompletely incorporated into the narrative. For example, the theoretical vaccine coverage BAST and MenDeVAR index is not described in M&M and it is not discussed in the discussion section what this information adds. Additionally, although very important, without a clearer aim it is unclear why the Czech isolates were compared to European isolates and then non-European isolates.

These recommendations were accepted – please see in the enclosure „Revised Manuscript with Track Changes“.

2. The results section lacks a clear narrative and is difficult to get through. Although there is no limit on word count, the manuscript is 6000 words long and seems to be unbalanced, 3 pages of introduction, 13 pages of results, 3 pages of discussion. The results section would benefit from being shortened and some parts moved to the discussion.

These recommendations were accepted – please see in the enclosure „Revised Manuscript with Track Changes“.

3. Figure S1 gives the reader an important sense of how the serogroup Y incidence, although increasing, is still very low in the Czech Republic compared to serogroups B and C. The authors could argue that it is rather the high mortality rate of serogroup Y that is the issue but this needs to be clearer as the introduction focuses on the increased serogroup Y IMD incidence.

These recommendations were accepted – please see in the enclosure „Revised Manuscript with Track Changes“.

4. Everything is relative but calling the serogroup Y isolates heterogeneous when 2/3 belonged to cc23 and then looking at non-European isolates which were even more heterogeneous, maybe then one would say that the Czech isolates were quite homogeneous? This could be re-phrased to be a more “soft” statement throughout the manuscript.

This recommendation was accepted – please see in the enclosure „Revised Manuscript with Track Changes“.

5. The discussion is incomplete and mostly mentions other studies. The most important findings should be highlighted and how the results adds to the field of research, possible limitations with the study and future directions should be added.

These recommendations were accepted – please see in the enclosure „Revised Manuscript with Track Changes“.

6. The manuscript would benefit from a general language revision, especially in regard to grammatical tense.

This recommendation was accepted – please see in the enclosure „Revised Manuscript with Track Changes“.

Minor points:

All figures would benefit from having the N= written in the figure legends instead of in the main text. 

This recommendation was accepted – please see in the enclosure „Revised Manuscript with Track Changes“.

Abbreviations NRL, ST and cc in the legends should be explained.

Abbreviations NRL, ST and cc is explained in Materials and Methods – please see in the enclosure „Revised Manuscript with Track Changes“.

Line 20-22: The material and methods section of the abstract is somewhat short, it does not for example mention that the Czech isolates were compared to other countries.

This recommendation was accepted – please see in the enclosure „Revised Manuscript with Track Changes“.

Line 59: The cause of shifting meningococcal serogroup incidences in meningococci is not completely known and vaccination programmes is probably only part of the explanation, the sentence could be rephrased to e.g. “…partly due to vaccination programmes”.

This recommendation was accepted – please see in the enclosure „Revised Manuscript with Track Changes“.

Line 73: Please add a reference to this statement.

This recommendation was accepted – please see in the enclosure „Revised Manuscript with Track Changes“.

Line 78: The study was from 2015, not 2014.

This recommendation was accepted – please see in the enclosure „Revised Manuscript with Track Changes“.

Line 80: WGS was done in reference number 14.

This recommendation was accepted – please see in the enclosure „Revised Manuscript with Track Changes“.

Line 88: What is the timeline for the numbers on serogroup distribution, since 1993?

This recommendation was accepted – please see in the enclosure „Revised Manuscript with Track Changes“.

Introduction: please add numbers on IMD incidence in the Czech Republic. These would be interesting to see in conjunction to the vaccine strategy.

This recommendation was accepted – please see in the enclosure „Revised Manuscript with Track Changes“.

Line 128: There is no information about the healthy carriers, in which setting were these sampled? What was the purpose of these samples in the study?

This recommendation was accepted – please see in the enclosure „Revised Manuscript with Track Changes“.

Material and methods: The BAST, MenDeVAR index and rMLST are not mentioned.

This recommendation was accepted – please see in the enclosure „Revised Manuscript with Track Changes“.

Table 1: The entire table is not visible. The abbreviations “CAR” and “PAC” are not explained.

As explained above, unfortunately not all Table 1 columns were visible in the first version of the manuscript. In the revised version of the manuscript which we are now submitting, Table 1 was divided into Table 1 and Table 2 and all columns are visible. The column with the abbreviations CAR and PAC was deleted according to the recommendation of the academic editor. Isolates from IMD versus isolates from carriers are distinguished by bold in the column Strain IDs in Table 1 and Table 2. 

Line 215-216: Please give an estimation on how many the minimal genetic differences are.

This comment was accepted, the part of sentence about minimal genetic differences was deleted – please see in the enclosure „Revised Manuscript with Track Changes“.

Line 295-303: This should be mentioned in the Material and Methods section.

This recommendation was accepted – please see in the enclosure „Revised Manuscript with Track Changes“.

Line 479: Please use “n=3” instead of “3x ST-2880”, should be changed throughout the manuscript.

The expressions such as "3x" have been deleted from the text throughout the manuscript as part of the recommended shortening of the text.

Line 494-503: This type of reasoning is better fitted for the discussion section rather than the results.

This recommendation was accepted – please see in the enclosure „Revised Manuscript with Track Changes“.

Line 515: The discussion starts with comparisons of which STs dominates in different countries, suggestion to add that the YI strain in Sweden belongs to ST-23.

This recommendation was accepted – please see in the enclosure „Revised Manuscript with Track Changes“.

Discussion: In the introduction it is stated that molecular characterization of serogroup Y provides background data for vaccination strategy, how the results from the current study will impact vaccination strategies could be discussed.

This recommendation was accepted – please see in the enclosure „Revised Manuscript with Track Changes“.

Fig 3-4: The roman letters are not explained in the figure legend.

This recommendation was accepted – please see in the enclosure „Revised Manuscript with Track Changes“.

Response to the Reviewer #3: 

Your work represents a nice work and adds valuable information of the epidemiologically situation based on whole genome sequencing of Neisseria meningitides serogroup Y isolates, collected in the Czech Republic during 1993-2018, and also its relation to other international isolates. The manuscript though needs extensive revisions, especially regarding the structure and focus of the scope of aim, i.e all parts regarding vaccination need to be shortened or in some places even taken away.

These recommendations were accepted – please see in the enclosure „Revised Manuscript with Track Changes“.

You can see more in detail below.

1. A specified aim is missing both in the manuscript and in the abstract.

This recommendation was accepted – please see in the enclosure „Revised Manuscript with Track Changes“.

2. The introduction is too long and need to be both structured and written in a logic way. For example, the section about vaccination is to detailed and needs to be shortened substantially, it should also be mentioned that the incidence and variation of causative serogroups varies over time not only due to vaccination but also natural fluctuations.

These recommendations were accepted – please see in the enclosure „Revised Manuscript with Track Changes“.

3. In M&M the collection of carrier isolates need to be described how it was done.

This recommendation was accepted – please see in the enclosure „Revised Manuscript with Track Changes“.

4. All subheadings (in the entire manuscript) need to be optimised and shortened, i.e “DNA extraction, WGS protocol, and WGS data analysis and visualization” could be Whole genome sequencing and analysis or even divided into sections.

This recommendation was accepted – please see in the enclosure „Revised Manuscript with Track Changes“.

5. The technically sound of the sequencing must be improved.

We do not understand this recommendation. The sequencing used in the present study have been described in detail previously, as is mentioned in M&M part [ref. 17 and 18]. 

6. All abbreviations in the Tables and figures need to be explained in footnotes or used without abbreviation and print full name, especially in the headings (should be possible to read stand-alone).

This recommendation was accepted – please see in the enclosure „Revised Manuscript with Track Changes“.

7. The result section is too long and should only include the actual results and not new background information or interpretations. Do not repeat all results that can be found in tables and figures, just the most important findings.

This recommendation was accepted – please see in the enclosure „Revised Manuscript with Track Changes“.

8. The interpretations can be moved to the section of discussion. Also do not include all results from every country in the discussion only the comparison.

This recommendation was accepted – please see in the enclosure „Revised Manuscript with Track Changes“.

9. The conclusion (both in manuscript and abstract) needs to answer an aim (that first has to be defined…).

This recommendation was accepted – please see in the enclosure „Revised Manuscript with Track Changes“.

10. All figures need better solution.

When submitting the corrected version of the manuscript, we have uploaded our figure files to the Preflight Analysis and Conversion Engine (PACE) digital diagnostic tool https://pacev2.apexcovantage.com/ to be sure that our figures meet PLOS requirements.

---

## [Decision Letter · Decision Letter 1]

3 Jan 2022

PONE-D-21-29616R1Whole genome sequencing of Neisseria meningitidis Y isolates collected in the Czech Republic in 1993-2018PLOS ONE

Dear Dr. Krizova,

Thank you for submitting your manuscript to PLOS ONE. After careful consideration, we feel that it has merit but does not fully meet PLOS ONE’s publication criteria as it currently stands. Therefore, we invite you to submit a revised version of the manuscript that addresses the points raised during the review process.

Both reviewers agreed that the revised manuscript showed substantial improvement, however, some new points have arisen and need to be addressed carefully.  Please see reviewers’ insightful comments below.  Also, there are numerous issues where additional clarification is needed (specific comments). 

In addition, the quality of the language still needs to be improved.  We suggest you thoroughly copyedit your manuscript for language usage, spelling, and grammar. If you do not know anyone who can help you do this, you may wish to consider employing a professional scientific editing service.

Specific comments:

Line 193 – 195:  The statement “Analysis of the theoretical…” is not relevant to the manuscript, suggest deletion.Line 215:  Suggest adding the following statement "The highest incidence of IMD caused by *N. meningitidis* Y in Czech Republic is in adolescents/young adults  (age 15 - 19), followed by older adults (≥65 years of age) and in younger children (1 - 4 years old). (Fig. S3)”  Suggest changing to Fig. S3 to S2 in order to reflect the fact that Fig. S3 appeared first.Line 249:  Suggest adding the following statement “The highest incidence of IMD caused by *N. meningitidis* Y in Czech Republic is in adolescents/young adults  (age 15 - 19), followed by older adults (≥65 years of age) and in younger children (1 - 4 years old)”Line 277:  Are you referred to fig.2?  If so, please mentioned here.Line 277:  Suggest changing “A more detailed representation shows…” to “A more detailed phylogenetic analysis of cc23 isolates (Fig. 2) showed…”Line 334:  It will also be nice to break out how many isolates are from 1986 - 1999, 2000 - 2009, 2010 - 2015, 2016 – 2020.Line 336 – 343:  Combine Fig. 3A and 3B legend into one.  **Fig 3. Genetic diversity of *N. meningitidis *Y invasive isolates from the Czech Republic and European countries. **A) Genetic diversity of *N. meningitidis *Y isolates collected from 1986 to 2020:  Isolates (n = 995) are coloured according to detection year. Invasive isolates from Czech Republic (n = 35) are marked with a dark blue square and labelled by their NRL number, ST, and cc. B) Genetic relationship of *N. meningitidis *Y isolates based on geographic origins:  Isolates (n = 995) are coloured according to the country of origin. The individual lineages of the cc23 clonal complex are marked by Roman numerals (I – IX).  In addition, suggest including similar coloring scheme for Fig. 3B.Line 352 – 354:  Fig. 3A seems to show that cc23 were more prevalent/diversified from 2010 - 2020.  Only one from 1986 - 1999 and few from 2000 - 2010 which does not collaborate the statement here. Please clarify.Line 354:  Change “…all time period” to “…all time period (Fig. 3B)”.  Also suggest using similar coloring scheme for Fig. 3B.Line 402:  Similarly, it will also be nice to break out how many isolates are from 1993 - 1999, 2000 - 2009, 2010 - 2015, 2016 – 2020.Line 404 – 413:  Similarly, please combine Fig. 4A and 4B legend into one. **Fig 4. Genetic diversity of *N. meningitidis *Y invasive isolates from the Czech Republic**
**and non-European countries. **A) Genetic diversity of *N. meningitidis *Y isolates collected from 1993 to 2020:  Isolates (n = 410) are coloured according to detection year. Invasive isolates from Czech Republic (n = 35) are marked with a dark blue square and labelled by their NRL number, ST, and cc. B) Genetic relationship of *N. meningitidis *Y isolates based on geographic origins:  Isolates (n = 410) are coloured according to the country of origin. The individual lineages of the cc23 clonal complex are marked by Roman numerals (I – IV). Invasive isolates from Czech Republic (n =35) are marked with a dark blue square and labelled by their NRL number, ST, and cc.  Also, Fig.4B should retain the same coloring as Fig. 4A as well.Line 514:  Why discuss the cc23 in Sweden?  Does this correlate with Czech?Discussion:  Length discussion of cc23 in Sweden, England, Wales, USA, and Canada. Not sure what the authors wish to convey.  Suggest consolidating these sections and shorten the discussion.

We look forward to receiving your revised manuscript.

Kind regards,

Baochuan Lin, Ph.D.

Academic Editor

PLOS ONE

Reviewers' comments:

Reviewer's Responses to Questions

**Comments to the Author**

1. If the authors have adequately addressed your comments raised in a previous round of review and you feel that this manuscript is now acceptable for publication, you may indicate that here to bypass the “Comments to the Author” section, enter your conflict of interest statement in the “Confidential to Editor” section, and submit your "Accept" recommendation.

Reviewer #1: (No Response)

Reviewer #2: All comments have been addressed

2. Is the manuscript technically sound, and do the data support the conclusions?

Reviewer #1: Partly

Reviewer #2: (No Response)

3. Has the statistical analysis been performed appropriately and rigorously? 

Reviewer #1: N/A

Reviewer #2: (No Response)

4. Have the authors made all data underlying the findings in their manuscript fully available?

Reviewer #1: Yes

Reviewer #2: (No Response)

5. Is the manuscript presented in an intelligible fashion and written in standard English?

Reviewer #1: No

Reviewer #2: (No Response)

6. Review Comments to the Author

Reviewer #1: Please see attached comments and recommendations

I want to thank the authors for the interesting research and for the meticulous detailed points mentioned in the manuscript.I have few points some are recommendations and others are inquiries .

This manuscript has been re-written and reads much better as suggested by the previous reviewers . However further editing for clarity would be advised.

Abstract:

Line 38-39: The WGS has been described previously (This Phrase can be removed and mentioned in the M&M)

Any mention about studying the relatedness to vaccine strains in the abstract ?whether in the aim ,results ?

Introduction:

Line 105-109

The Swedish isolates comment can be removed from the introduction as you already mentioned it in the discussion

Methods

Paragraph from line 208 to 226 : would you clarify why you choose Vaccine covering group B ,you mentioned there are some protection but why we are studying vaccine covering group B and you have already vaccine covering Y strains why these vaccines were not studied ? would you elaborate more in this point .

Results:

Paragraph starting by line 269,these results of similarities between vaccines for Men B and the studied isolates, what is the interpretation of these results ,are you going to recommend these vaccines instead of the current vaccines for Y strains ?what is your recommendation after mentioning the color coding similarities

Is it possible to mention this in the discussion part and how these results will update the vaccine strategy .

Line 347: would you mention at the beginning how many invasive and how many carriers having the CC92 and the same for CC23

Too much details for the infrequent CC ,better to summarize this part only mentioning the main findings.

Fig3A and 3B have the same title ?

Paragraph started by line 495: Too much details ,you can summarize this paragraph as this is the infrequent non CC23,you can put all non 23CC in one paragraph.

Paragraph started by line 567: summarize lineage ii the main findings you cannot describe each and every sub-lineage ,only general description because everything is shown in the figure so you mention what you cannot describe in the figure

Discussion :

Line 648 : I think this was mentioned before by one of the reviewer that if 2/3 of isolates are CC23 how can we considered that as heterogeneous?

Line657: The non-European population of isolates of N. meningitidis Y appears to be less diverse or more diverse

Line 670 to 679 would you rephrase the sentence please, the meaning is not clear (the whole paragraph need to be rewritten )

Line 744 to 748: how these results will update on the strategy of vaccine coverage in Czech Republic.

Reviewer #2: The authors have made substantial changes to the manuscript and many of my points have been addressed, however, some new points have arisen:

(the lines refer to the version with tracked changes)

Lines 218-223, it is a bit confusing listing the colors according to MenDeVar index here without a reference to a table. The colors could be left out in the material and method section and clarified in the table instead.

Line 723-724, it is unclear why cc175 is mentioned in this paragraph, is this not related to the previous paragraph and should be moved to line 717?

7. PLOS authors have the option to publish the peer review history of their article (what does this mean?). If published, this will include your full peer review and any attached files.

Reviewer #1: **Yes: **Manal Mahmoud Hamed

Reviewer #2: No

---

## [Author Response · Author response to Decision Letter 1]

20 Jan 2022

PONE-D-21-29616R1

Whole genome sequencing of Neisseria meningitidis Y isolates collected in the Czech Republic in 1993-2018

PLOS ONE

The authors thank the academic editor and two reviewers for their valuable comments and recommendations. Most of them were accepted - see the answers to the points below.

Response to the academic editor:

Both reviewers agreed that the revised manuscript showed substantial improvement, however, some new points have arisen and need to be addressed carefully. Please see reviewers’ insightful comments below. Also, there are numerous issues where additional clarification is needed (specific comments). 

These recommendations were mostly accepted – please see in the enclosure „Revised Manuscript with Track Changes“. 

In addition, the quality of the language still needs to be improved. We suggest you thoroughly copyedit your manuscript for language usage, spelling, and grammar. If you do not know anyone who can help you do this, you may wish to consider employing a professional scientific editing service.

The English proofing of the second revised version of the manuscript was carried out by Keith Jolley of the University of Oxford, who, in addition to being a native speaker, is also a world-renowned expert in whole genome sequencing of Neisseria meningitidis.

Specific comments:

1. Line 193 – 195: The statement “Analysis of the theoretical…” is not relevant to the manuscript, suggest deletion.

This recommendation was accepted – please see in the enclosure „Revised Manuscript with Track Changes“. 

2. Line 215: Suggest adding the following statement "The highest incidence of IMD caused by N. meningitidis Y in Czech Republic is in adolescents/young adults (age 15 - 19), followed by older adults (≥65 years of age) and in younger children (1 - 4 years old). (Fig. S3)” Suggest changing to Fig. S3 to S2 in order to reflect the fact that Fig. S3 appeared first.

This recommendation was accepted – please see in the enclosure „Revised Manuscript with Track Changes“. 

3. Line 249: Suggest adding the following statement “The highest incidence of IMD caused by N. meningitidis Y in Czech Republic is in adolescents/young adults (age 15 - 19), followed by older adults (≥65 years of age) and in younger children (1 - 4 years old)”

This information would already be duplicated, given that we have already added the age information above, based on a previous recommendation. 

4. Line 277: Are you referred to fig.2? If so, please mentioned here.

This recommendation was accepted – please see in the enclosure „Revised Manuscript with Track Changes“. 

5. Line 277: Suggest changing “A more detailed representation shows…” to “A more detailed phylogenetic analysis of cc23 isolates (Fig. 2) showed…”

This recommendation was accepted – please see in the enclosure „Revised Manuscript with Track Changes“. 

6. Line 334: It will also be nice to break out how many isolates are from 1986 - 1999, 2000 - 2009, 2010 - 2015, 2016 – 2020.

This recommendation was accepted – please see in the enclosure „Revised Manuscript with Track Changes“. 

7. Line 336 – 343: Combine Fig. 3A and 3B legend into one. Fig 3. Genetic diversity of N. meningitidis Y invasive isolates from the Czech Republic and European countries. A) Genetic diversity of N. meningitidis Y isolates collected from 1986 to 2020: Isolates (n = 995) are coloured according to detection year. Invasive isolates from Czech Republic (n = 35) are marked with a dark blue square and labelled by their NRL number, ST, and cc. B) Genetic relationship of N. meningitidis Y isolates based on geographic origins: Isolates (n = 995) are coloured according to the country of origin. The individual lineages of the cc23 clonal complex are marked by Roman numerals (I – IX). In addition, suggest including similar coloring scheme for Fig. 3B.

This recommendation was accepted – please see in the enclosure „Revised Manuscript with Track Changes“. We do not understand the additional suggestion to include similar colors in Fig. 3B, because Fig. 3A presents different periods while Fig. 3B presents different countries.

8. Line 352 – 354: Fig. 3A seems to show that cc23 were more prevalent/diversified from 2010 - 2020. Only one from 1986 - 1999 and few from 2000 - 2010 which does not collaborate the statement here. Please clarify.

This recommendation was accepted – please see in the enclosure „Revised Manuscript with Track Changes“. 

9. Line 354: Change “…all time period” to “…all time period (Fig. 3B)”. Also suggest using similar coloring scheme for Fig. 3B.

This recommendation was accepted – please see in the enclosure „Revised Manuscript with Track Changes“. We do not understand the additional suggestion to include similar colors in Fig. 3B, because Fig. 3A presents different periods while Fig. 3B presents different countries.

10. Line 402: Similarly, it will also be nice to break out how many isolates are from 1993 - 1999, 2000 - 2009, 2010 - 2015, 2016 – 2020.

This recommendation was accepted – please see in the enclosure „Revised Manuscript with Track Changes“. 

11. Line 404 – 413: Similarly, please combine Fig. 4A and 4B legend into one. Fig 4. Genetic diversity of N. meningitidis Y invasive isolates from the Czech Republic and non-European countries. A) Genetic diversity of N. meningitidis Y isolates collected from 1993 to 2020: Isolates (n = 410) are coloured according to detection year. Invasive isolates from Czech Republic (n = 35) are marked with a dark blue square and labelled by their NRL number, ST, and cc. B) Genetic relationship of N. meningitidis Y isolates based on geographic origins: Isolates (n = 410) are coloured according to the country of origin. The individual lineages of the cc23 clonal complex are marked by Roman numerals (I – IV). Invasive isolates from Czech Republic (n =35) are marked with a dark blue square and labelled by their NRL number, ST, and cc. Also, Fig.4B should retain the same coloring as Fig. 4A as well.

This recommendation was accepted – please see in the enclosure „Revised Manuscript with Track Changes“. We do not understand the additional suggestion to include similar colors in Fig. 4B, because Fig. 4A presents different periods while Fig. 4B presents different countries.

12. Line 514: Why discuss the cc23 in Sweden? Does this correlate with Czech?

This recommendation was accepted – please see in the enclosure „Revised Manuscript with Track Changes“. 

13. Discussion: Length discussion of cc23 in Sweden, England, Wales, USA, and Canada. Not sure what the authors wish to convey. Suggest consolidating these sections and shorten the discussion.

This recommendation was accepted – please see in the enclosure „Revised Manuscript with Track Changes“. 

Response to the Reviewer #1:

Abstract:

Line 38-39: The WGS has been described previously (This Phrase can be removed and mentioned in the M&M)

Any mention about studying the relatedness to vaccine strains in the abstract ?whether in the aim ,results ?

This recommendation was accepted – please see in the enclosure „Revised Manuscript with Track Changes“. 

Introduction:

Line 105-109

The Swedish isolates comment can be removed from the introduction as you already mentioned it in the discussion

This recommendation was accepted – please see in the enclosure „Revised Manuscript with Track Changes“. 

Methods

Paragraph from line 208 to 226 : would you clarify why you choose Vaccine covering group B ,you mentioned there are some protection but why we are studying vaccine covering group B and you have already vaccine covering Y strains why these vaccines were not studied ? would you elaborate more in this point .

This recommendation was accepted – please see in the enclosure „Revised Manuscript with Track Changes“. 

Results:

Paragraph starting by line 269,these results of similarities between vaccines for Men B and the studied isolates, what is the interpretation of these results ,are you going to recommend these vaccines instead of the current vaccines for Y strains ? what is your recommendation after mentioning the color coding similarities

This recommendation was accepted – please see in the enclosure „Revised Manuscript with Track Changes“. 

Is it possible to mention this in the discussion part and how these results will update the vaccine strategy .

This was already stated in the previous version of the manuscript: …in the Czech Republic the conjugated tetravaccine ACYW started to be used instead of the previous conjugated monovaccine C. - so we leave it unchanged.

Line 347: would you mention at the beginning how many invasive and how many carriers having the CC92 and the same for CC23

This was already stated in the previous version for cc92 - so we are adding it also for cc23 – please see in the enclosure „Revised Manuscript with Track Changes“. 

Too much details for the infrequent CC ,better to summarize this part only mentioning the main findings.

This recommendation was accepted – please see in the enclosure „Revised Manuscript with Track Changes“. 

Fig3A and 3B have the same title ?

This recommendation is similar to the recommendation of the academic editor (see above) and was accepted: the title of Fig. 3A and 3B has been merged together – please see in the enclosure „Revised Manuscript with Track Changes“. 

Paragraph started by line 495: Too much details ,you can summarize this paragraph as this is the infrequent non CC23,you can put all non 23CC in one paragraph.

This recommendation was accepted – please see in the enclosure „Revised Manuscript with Track Changes“. 

Paragraph started by line 567: summarize lineage ii the main findings you cannot describe each and every sub-lineage ,only general description because everything is shown in the figure so you mention what you cannot describe in the figure

This recommendation was accepted – please see in the enclosure „Revised Manuscript with Track Changes“. 

Discussion :

Line 648 : I think this was mentioned before by one of the reviewer that if 2/3 of isolates are CC23 how can we considered that as heterogeneous?

This recommendation was accepted – please see in the enclosure „Revised Manuscript with Track Changes“. 

Line657: The non-European population of isolates of N. meningitidis Y appears to be less diverse or more diverse

It was already stated in the previous version of the manuscript that the non-European population appears to be less diverse - so we leave it unchanged. 

Line 670 to 679 would you rephrase the sentence please, the meaning is not clear (the whole paragraph need to be rewritten )

This recommendation was accepted – please see in the enclosure „Revised Manuscript with Track Changes“. 

Line 744 to 748: how these results will update on the strategy of vaccine coverage in Czech Republic.

This was already stated in the previous version of the manuscript: …in the Czech Republic the conjugated tetravaccine ACYW started to be used instead of the previous conjugated monovaccine C. - so we leave it unchanged.

Response to the Reviewer #2:

Lines 218-223, it is a bit confusing listing the colors according to MenDeVar index here without a reference to a table. The colors could be left out in the material and method section and clarified in the table instead.

This recommendation was accepted – please see in the enclosure „Revised Manuscript with Track Changes“. 

Line 723-724, it is unclear why cc175 is mentioned in this paragraph, is this not related to the previous paragraph and should be moved to line 717?

This recommendation was accepted – please see in the enclosure „Revised Manuscript with Track Changes“.

---

## [Decision Letter · Decision Letter 2]

23 Feb 2022

Whole genome sequencing of Neisseria meningitidis Y isolates collected in the Czech Republic in 1993-2018

PONE-D-21-29616R2

Dear Dr. Krizova,

We’re pleased to inform you that your manuscript has been judged scientifically suitable for publication and will be formally accepted for publication once you corrected the following errors and it meets all outstanding technical requirements. 1) Line 205 - 206, (ST-23, n = 7; ST-569, n = 4; ST-1625, n = 6; and ST-8526, n = 1). These numbers do not add up to 21 isolates, please correct.  2) Line 348, extra period at the end of the sentence, please correct.

Kind regards,

Baochuan Lin, Ph.D.

Academic Editor

PLOS ONE

Additional Editor Comments (optional):

Reviewers' comments:

Reviewer's Responses to Questions

**Comments to the Author**

1. If the authors have adequately addressed your comments raised in a previous round of review and you feel that this manuscript is now acceptable for publication, you may indicate that here to bypass the “Comments to the Author” section, enter your conflict of interest statement in the “Confidential to Editor” section, and submit your "Accept" recommendation.

Reviewer #2: (No Response)

2. Is the manuscript technically sound, and do the data support the conclusions?

Reviewer #2: (No Response)

3. Has the statistical analysis been performed appropriately and rigorously? 

Reviewer #2: (No Response)

4. Have the authors made all data underlying the findings in their manuscript fully available?

Reviewer #2: (No Response)

5. Is the manuscript presented in an intelligible fashion and written in standard English?

Reviewer #2: (No Response)

6. Review Comments to the Author

Reviewer #2: (No Response)

7. PLOS authors have the option to publish the peer review history of their article (what does this mean?). If published, this will include your full peer review and any attached files.

Reviewer #2: No

---

## [Editor Report · Acceptance letter]

2 Mar 2022

PONE-D-21-29616R2 

Whole genome sequencing of *Neisseria meningitidis* Y isolates collected in the Czech Republic in 1993-2018 

Dear Dr. Krizova:

I'm pleased to inform you that your manuscript has been deemed suitable for publication in PLOS ONE. Congratulations! Your manuscript is now with our production department. 

Kind regards, 

on behalf of

Dr. Baochuan Lin 

Academic Editor

PLOS ONE